# THEORETICAL INSIGHTS INTO FINE-TUNING ATTENTION MECHANISM: GENERALIZATION AND OPTIMIZATION

## ABSTRACT

Large Language Models (LLMs), built on Transformer architectures, exhibit remarkable generalization across a wide range of tasks. However, fine-tuning these models for specific tasks remains resource-intensive due to their extensive parameterization. In this paper, we investigate two remarkable phenomena related to the attention mechanism during the fine-tuning of LLMs. The first phenomenon, termed "Unequal Importance of Attention Matrices," highlights the impact of fine-tuning different weight matrices. It shows that optimizing the $\mathbf{W}_v$ matrix yields significantly better performance than optimizing the $\mathbf{W}_k$ matrix. Fine-tuning only the $\mathbf{W}_q$ and $\mathbf{W}_v$ matrices is computationally efficient while delivering results comparable to, or even better than fine-tuning all three matrices ($\mathbf{W}_q$, $\mathbf{W}_k$, and $\mathbf{W}_v$). The second phenomenon, "Attention Matrices with Customized Learning Rate Leads to Better Convergence," emphasizes the importance of assigning distinct learning rates to these matrices. Specifically, a higher learning rate for the $\mathbf{W}_v$ matrix compared to $\mathbf{W}_q$ and $\mathbf{W}_k$ accelerates convergence and improves performance. Building on these insights, we propose a new strategy that improves fine-tuning efficiency in terms of both storage and time. Experimental results on benchmark datasets validate the effectiveness of this approach, supporting our theoretical findings. Our analysis lays the theoretical groundwork for configuring and improving lightweight algorithms in LLMs fine-tuning.

## 1 INTRODUCTION

Large Language Models (LLMs) are often built on Transformer architectures [43] and possess a large number of parameters, enabling them to generalize across a broad range of general tasks [43, 27, 42, 8, 31]. However, achieving optimal performance on specific tasks typically necessitates fine-tuning these pre-trained models. Despite the formidable capabilities of LLMs, the fine-tuning process is resource-intensive, requiring significant computational power, storage, and time due to the large scale of model parameters involved. Fine-tuning all the parameters of a large language model, known as full fine-tuning, is highly computationally expensive. To reduce the computational cost, various parameter-efficient fine-tuning (PEFT) methods have been proposed [7, 19, 23, 24, 20], which only fine-tune a small number of (extra) model parameters.

A fundamental component of transformers is the attention mechanism, particularly the interactions among the query matrix $\mathbf{W}_q$, the key matrix $\mathbf{W}_k$, and the value matrix $\mathbf{W}_v$. During the fine-tuning of LLMs involving the attention mechanism, two interesting phenomena have been observed: (1) *Unequal Importance of Attention Matrices*—optimizing the $\mathbf{W}_v$ matrix is pivotal for enhancing performance, significantly more so than adjustments to the $\mathbf{W}_k$ matrix, which exhibit limited impact on the outcomes. Additionally, fine-tuning only the $\mathbf{W}_q$ and $\mathbf{W}_v$ matrices often yields results that are comparable to or surpass those achieved by fine-tuning all three matrices $\mathbf{W}_q$, $\mathbf{W}_k$, and $\mathbf{W}_v$, which also reduces the number of tunable parameters by approximately 1/3, offering computational benefits (Section 3). (2) *Attention Matrices with Customized Learning Rate Leads to Better Convergence*—using the same learning rate for $\mathbf{W}_q$&$\mathbf{W}_k$ and $\mathbf{W}_v$ is not optimal for efficient convergence. In fact, it is essential to apply distinct learning rates for the $\mathbf{W}_q$, $\mathbf{W}_k$, and $\mathbf{W}_v$ components to ensure optimal fine-tuning performance. Specifically, the learning rate for $\mathbf{W}_v$ should generally be higher than that for $\mathbf{W}_q$ and $\mathbf{W}_k$ to facilitate efficient convergence (Section 4).

While certain empirical guidelines, such as the original Low-Rank Adaptation (LoRA) [20], explore which weight matrices in transformers are suitable for the application of LoRA, comprehensive theoretical analyses of these phenomena are still limited. This includes aspects such as selecting appropriate weight types for fine-tuning and optimizing learning rate settings. Reflecting on the attention equation itself (Section 2): (1) In linear algebra, two matrices multiplied without an intermediate activation can be equivalent to a single matrix. Some studies [33, 40, 4] often treat $\mathbf{W}_q$ and $\mathbf{W}_k$ as a single unit ($\mathbf{W}_{qk} = \mathbf{W}_q \mathbf{W}_k^T$), however, the benefits of fine-tuning $\mathbf{W}_q \& \mathbf{W}_v$ alone have yet to be further clarified. (2) Cosidering the scenario where the values of $\mathbf{W}_q$, $\mathbf{W}_k$, and $\mathbf{W}_v$ approach zero, the gradients of $\mathbf{W}_q \& \mathbf{W}_k$ tend to diminish towards zero. In contrast, the gradient of $\mathbf{W}_v$ remains non-zero due to the influence of softmax normalization. Driven by the above motivations, this paper delves into the issue from the following two perspectives.

- **Generalization: advantages of fine-tuning $\mathbf{W}_q \& \mathbf{W}_v$ over $\mathbf{W}_q, \mathbf{W}_k, \mathbf{W}_v$ together.** We perform a thorough theoretical analysis to demonstrate the advantages. To be more specific, we employ information-theoretic approaches [46, 34, 45, 53] to establish the generalization bounds of fine-tuning pre-trained models with attention mechanism (See **Theorem 1** for details). This indicates that fine-tuning $\mathbf{W}_q \& \mathbf{W}_v$ instead of $\mathbf{W}_q, \mathbf{W}_k, \mathbf{W}_v$ reduces the number of parameters, while improving generalization bounds and potentially providing memory benefits.

- **Optimization: convergence analysis of attention mechanism with varying learning rate settings.** To further investigate the aforementioned phenomena, we examine the optimization process of the attention mechanism. First, we discuss the learning dynamics in transformers in **Case 1**, suggest that $\mathbf{W}_v$ may experience instances of inefficient learning during fine-tuning process for downstream tasks. This naturally leads to the hypothesis that accelerating the learning of $\mathbf{W}_v$ in the early stages could potentially induce $\mathbf{W}_k$ and $\mathbf{W}_q$ to begin learning earlier. Additionally, by using scaling arguments for large width-$n$ networks [49, 13], we illustrate (**Theorem 2**) that the feature learning of attention mechanism is efficient when the learning rate for $\mathbf{W}_v$ should be generally much larger than that of $\mathbf{W}_q \& \mathbf{W}_k$ in fine-tuning.

Building on our experimental and theoretical insights, one can develop new algorithms to improve the effectiveness (e.g., storage, and time) of fine-tuning. Experimental results for our strategy (in Section 5) on benchmark datasets [44] and open source pre-trained models [29, 2] verify that the method can visibly influence fine-tuning efficiency[1]. We do not make direct comparisons with various parameter-efficient fine-tuning methods, as our strategy is primarily intended to demonstrate how theoretical analysis can effectively guide experimental procedures.

## 2 PRELIMINARIES AND BACKGROUND

In this section, we first describe the core components of our study by reviewing some basic notations. The transformer model [43] serves as the backbone of most state-of-the-art pre-trained models. For clarity, we briefly outline its key equations, focusing on the self-attention function, as follows.

**Self-attention.** Given a sequence of $m$ vectors $\mathbf{C} \in \mathbb{R}^{m \times d_{in}}$ over which we would like to perform attention and a query vector $\mathbf{x} \in \mathbb{R}^{d_{in}}$, that is, the input is $[\mathbf{C}, \mathbf{x}] \in \mathbb{R}^{(m+1) \times d_{in}}$. The conventional attention function can be expressed as[2]:

$$\text{Attn}(\mathbf{x}\mathbf{W}_q, \mathbf{C}\mathbf{W}_k, \mathbf{C}\mathbf{W}_v) = \text{softmax}\left( \frac{\mathbf{x}\mathbf{W}_q \mathbf{W}_k^T \mathbf{C}^T}{\sqrt{d_{out}}} \right) \mathbf{C}\mathbf{W}_v, \quad (1)$$

where $\mathbf{W}_q, \mathbf{W}_k, \mathbf{W}_v \in \mathbb{R}^{d_{in} \times d_{out}}$ are query, key and value (projection) matrices.

**A unified framework for parameter-efficient fine-tuning.** Building on the work of [16], we consider a unified framework that establishes connections among various parameter-efficient fine-tuning methods. Specifically, we reinterpret these methods as modifications applied to specific hidden states

---

[1]Code is anonymized at https://anonymous.4open.science/r/LightweightAtt-6899/
[2]For simplicity, we focus on the last vector of input in a single-head self-attention. Our analysis is readily generalizable to multi-head self-attention.

within pre-trained models, the composition function:

$$\mathbf{h} \leftarrow l_1 \mathbf{h} + l_2 \Delta \mathbf{h}, \tag{2}$$

where $l_1, l_2$ are coefficients, $\mathbf{h}$ is denoted as the hidden representation to be directly modified and $\Delta \mathbf{h}$ is a modification vector. Additionally, $\mathbf{h}$ and $\mathbf{x}$ can represent the attention output and input respectively. Here, we will present two special cases:

- **LoRA.** LoRA [20] injects trainable low-rank matrices into transformer layers to approximate the weight updates. Instead of directly adjusting the full weight matrix $\mathbf{W} \in \mathbb{R}^{d_{in} \times d_{out}}$, LoRA represents its update with a low-rank decomposition $\mathbf{W} + \Delta \mathbf{W} = \mathbf{W} + \mathbf{A}\mathbf{B}$, where $\mathbf{A} \in \mathbb{R}^{d_{in} \times r}, \mathbf{B} \in \mathbb{R}^{r \times d_{out}}$ are tunable parameters. For a specific input $\mathbf{x}$, LoRA modifies the projection output $\mathbf{h}$ as (where $s \geq 1$ is a tunable scalar hyper-parameter):

$$\mathbf{h} \leftarrow \mathbf{h} + s\Delta \mathbf{h}, \quad \Delta \mathbf{h} := \mathbf{x}\mathbf{A}\mathbf{B}. \tag{3}$$

- **Prefix tuning.** Prefix tuning [24] prepends $r$ tunable prefix vectors to the keys and values of the attention mechanism at every layer. Specifically, two sets of prefix vectors $\mathbf{P}_k, \mathbf{P}_v \in \mathbb{R}^{r \times d_{out}}$ are concatenated with the original key $\mathbf{C}\mathbf{W}_k$ and value $\mathbf{C}\mathbf{W}_v$, attention is then applied to the prefixed keys and values as[3]:

$$\mathbf{h} \leftarrow (1 - \lambda(\mathbf{x})\mathbf{h} + \lambda(\mathbf{x})\Delta \mathbf{h}, \quad \Delta \mathbf{h} := \text{softmax}(\mathbf{x}\mathbf{W}_q\mathbf{P}_k^T)\mathbf{P}_v \triangleq \text{softmax}(\mathbf{x}\mathbf{A})\mathbf{B}, \tag{4}$$

where $\lambda(\mathbf{x}) = \frac{\sum_i \exp(\mathbf{x}\mathbf{W}_q\mathbf{P}_k^T)_i}{\sum_i \exp(\mathbf{x}\mathbf{W}_q\mathbf{P}_k^T)_i + \sum_j \exp(\mathbf{x}\mathbf{W}_q\mathbf{W}_k^T\mathbf{C}^T)_j}$ is a scalar that represents the sum of normalized attention weights on the prefixes. We derive a detailed equivalent form of Prefix tuning to establish its connection with LoRA in Appendix B.1.

**Remark 1.** *By defining $\mathbf{A} = \mathbf{W}_q\mathbf{P}_k^T, \mathbf{B} = \mathbf{P}_v$ in Eq.(4), we can establish a connection with LoRA in Eq.(3). Notably, if we replace the softmax attention with linear attention here, the two are equivalent to some extent. Intuitively, in the attention mechanism, $\mathbf{A}$ ($\mathbf{W}_q\mathbf{P}_k^T$) is responsible for generating attention scores, while $\mathbf{B}$ ($\mathbf{P}_v$) utilizes these attention scores to produce the target content. Therefore, during fine-tuning, query, key, and value are likely to exhibit varying degrees of importance. This may also provide theoretical insights for recent works [53, 12], which empirically observed an asymmetry where the project-down matrix $\mathbf{A}$ is responsible for extracting features from the input, while the project-up matrix $\mathbf{B}$ utilizes these features to generate the desired output in LoRA fine-tuning.*

$\Theta$ **Notation.** For the convergence analysis, we adopt the following notation to describe the asymptotic behavior as the width $n$ increases, similar to those in [49, 13]. Given sequences $c_n \in \mathbb{R}$ and $d_n \in \mathbb{R}^+$, we write $c_n = O(d_n)$ and $c_n = \Omega(d_n)$ to mean $c_n < \kappa d_n$ or $c_n > \kappa d_n$, respectively, for some constant $\kappa > 0$. We denote $c_n = \Theta(d_n)$ when both $c_n = O(d_n)$ and $c_n = \Omega(d_n)$ hold, implying that $c_n$ and $d_n$ grow at comparable rates. For vector sequences $c_n = (c_n^i)_{1 \leq i \leq k} \in \mathbb{R}^k$ (for some $k > 0$), we write $c_n = O(d_n)$ when $c_n^i = O(d_n^i)$ for all $i \in [k]$, and analogous notation applies for other asymptotic bounds. Finally, when the sequence $c_n$ is a vector of random variables, convergence is understood to refer to convergence in the second moment (i.e., $L_2$ norm).

## 3 ADVANTAGES AND GENERALIZATION ANALYSIS

This section, we show our first interesting observation (*Unequal Importance of Attention Matrices*) in fine-tuning the attention mechanism and the storage benefit of fine-tuning only $\mathbf{W}_q$ and $\mathbf{W}_v$ in Section 3.1. Afterwards, we give a mutual information based generalization bounds of fine-tuning only $\mathbf{W}_q$ and $\mathbf{W}_v$ in Section 3.2, which provide a better generalization error.

### 3.1 EMPIRICAL ADVANTAGES OF FINE-TUNING ONLY QUERY, VALUE MATRICES

To explore the *Unequal Importance of Attention Matrices*, we focus our study on **adapting only the attention weights** for downstream tasks, while freezing the other modules to ensure simplicity

---

[3]Without loss of generalization, we ignore the softmax scaling factor for ease of notation.

and parameter efficiency. Furthermore, we investigate the impact of adapting different types of attention weight matrices in a Transformer, as outlined below. We present our empirical results using LoRA to fine-tune a set of language models (Roberta-base [29] and Llama3.1-8b [2]) across various benchmarks [44]. Further details on the experimental setup and additional empirical results can be found in Appendix C.1.

Table 1 provides a detailed comparison of the impact of fine-tuning different weight matrices $(\mathbf{W}_q, \mathbf{W}_k, \mathbf{W}_v)$ across various rank values $r$ and weight update strategies in LoRA fine-tuning on tasks like SST2, QNLI, QQP, and MNLI. As seen in the table, we can see a clear trend where solely updating the $\mathbf{W}_v$ matrix outperforms just learning the $\mathbf{W}_q, \mathbf{W}_k$ matrix. Interestingly, the combination of fine-tuning both $\mathbf{W}_q$ and $\mathbf{W}_v$ often leads to performance that matches or even exceeds that achieved by fine-tuning all three matrices $\mathbf{W}_q, \mathbf{W}_k$, and $\mathbf{W}_v$. This pattern is consistently observed across various tasks and rank values, further emphasizing the importance of these two matrices over $\mathbf{W}_k$ during fine-tuning.

**Computational benefits.** Here, we show that the reduced amount of adapted parameters by (roughly) $1/3$ provides computational gains. The key benefit of parameter-efficient method is to save memory during training, storage and communication [26]. Fine-tuning $\mathbf{W}_q \& \mathbf{W}_v$ alone as opposed to both $\mathbf{W}_q \& \mathbf{W}_v$ and $\mathbf{W}_k$ reduces the number of parameters by $1/3$, when the dimensions of $\mathbf{W}_q, \mathbf{W}_k$, and $\mathbf{W}_v$ are the same.

Table 1: Performance comparison across different $r$ values and weight types. To enable a fair comparison, we initialize the weights for all tasks with the original pretrained weights. Test accuracy of Roberta-base (R) and Llama3.1-8b (L) fine-tuning on SST2, QNLI, QQP, MNLI, with sequence length T = 128 and half precision (FP16). All values are averaged over 3 random seeds. The best result is shown in **bold**, the second best result is shown in underline, and the third best result is shown with double underlines.

| | Weight Type | $\mathbf{W}_q$ | $\mathbf{W}_k$ | $\mathbf{W}_v$ | $\mathbf{W}_q, \mathbf{W}_k$ | $\mathbf{W}_q, \mathbf{W}_v$ | $\mathbf{W}_q, \mathbf{W}_k, \mathbf{W}_v$ |
|---|---|---|---|---|---|---|---|
| **SST2**(R) | $r = 4$ | 0.904 | 0.902 | 0.913 | 0.919 | **0.920** | **0.920** |
| | $r = 8$ | 0.914 | 0.906 | 0.918 | 0.915 | 0.919 | **0.922** |
| | $r = 16$ | 0.907 | 0.905 | 0.916 | 0.917 | 0.921 | **0.923** |
| **QNLI**(R) | $r = 4$ | 0.854 | 0.835 | 0.878 | 0.866 | **0.888** | 0.887 |
| | $r = 8$ | 0.857 | 0.841 | 0.875 | 0.866 | 0.889 | **0.895** |
| | $r = 16$ | 0.854 | 0.840 | 0.875 | 0.867 | **0.890** | **0.890** |
| **QQP**(R) | $r = 4$ | 0.812 | 0.804 | 0.828 | 0.823 | 0.838 | **0.843** |
| | $r = 8$ | 0.812 | 0.806 | 0.828 | 0.823 | 0.840 | **0.844** |
| | $r = 16$ | 0.812 | 0.804 | 0.831 | 0.823 | 0.839 | **0.844** |
| **QQP**(L) | $r = 8$ | 0.864 | 0.845 | 0.865 | 0.866 | **0.874** | 0.874 |
| | $r = 16$ | 0.864 | 0.845 | 0.869 | 0.867 | **0.874** | 0.874 |
| **MNLI**(R) | $r = 4$ | 0.748 | 0.733 | 0.807 | 0.772 | 0.820 | **0.828** |
| | $r = 8$ | 0.749 | 0.733 | 0.809 | 0.778 | 0.820 | **0.827** |
| | $r = 16$ | 0.750 | 0.734 | 0.810 | 0.780 | 0.824 | **0.828** |
| **MNLI**(L) | $r = 8$ | 0.802 | 0.660 | 0.862 | 0.814 | **0.871** | 0.871 |
| | $r = 16$ | 0.803 | 0.663 | 0.863 | 0.815 | **0.871** | 0.871 |

**Why fine-tune $\mathbf{W}_q \& \mathbf{W}_v$ instead of $\mathbf{W}_k \& \mathbf{W}_v$.** In Eq.(1), the conventional attention function includes a term $\mathbf{x}\mathbf{W}_q\mathbf{W}_k^T$. (1) In linear algebra, two matrices multiplied without an intermediate activation can be equivalent to a single matrix. Therefore, the effects of fine-tuning $\mathbf{W}_q \& \mathbf{W}_v$ and $\mathbf{W}_k \& \mathbf{W}_v$ are theoretically expected to yield similar outcomes (See the supplementary experimental results provided in Appendix x). (2) $\mathbf{W}_k$ operates only on the transformed representation matrix $\mathbf{x}\mathbf{W}_q$ produced by the preceding transformation. Consequently, it loses direct access to the original representation information. This observation is consistent with the findings in [35]: the information

representation in LoRA also exhibits significant limitations, as in $\Delta \mathbf{h} = \mathbf{xAB}$, where $\mathbf{B}$ similarly lacks access to the original representation information.

## 3.2 INFORMATION-THEORETIC GENERALIZATION BOUNDS

In the previous part, we establish that the *Unequal Importance of Attention Matrices* among $\mathbf{W}_q$, $\mathbf{W}_k$, and $\mathbf{W}_v$ during fine-tuning. Some studies [33, 40, 4] often treat $\mathbf{W}_q$ and $\mathbf{W}_k$ as a single unit ($\mathbf{W}_{qk} = \mathbf{W}_q \mathbf{W}_k^T$), however, the benefits of fine-tuning $\mathbf{W}_q \& \mathbf{W}_v$ alone, rather than fine-tuning $\mathbf{W}_q \& \mathbf{W}_v$, and $\mathbf{W}_k$ together, have yet to be further clarified. Therefore, we will further analyze this issue from an information-theoretic generalization perspective.

Recently, information-theoretic generalization bounds [46, 37, 39, 45] have been introduced to analyze the expected generalization error of learning algorithms. A key benefit of these bounds is that they depend not only on the data distribution but also on the specific algorithm, making them an ideal tool for studying the generalization behavior of models trained using particular algorithms.

**Generalization error.** We let $\mathcal{Z} = \mathcal{X} \times \mathcal{Y}$ be the instance space and $\mu$ be an unknown distribution on $\mathcal{Z}$, specifying random variable $Z$. Here, $\mathcal{X}$ denotes the feature space and $\mathcal{Y}$ is the label space. Suppose one observes a training set $S_N \triangleq (Z_1, ..., Z_N) \in \mathcal{Z}^N$, with $N$ i.i.d. training examples drawn from $\mu$. In the information-theoretic analysis framework, we let $\mathcal{W}$ be the space of hypotheses related to the model, and a stochastic learning algorithm $\mathcal{A}$ which takes the training examples $S_N$ as its input and outputs a hypothesis $W \in \mathcal{W}$ according to some conditional distribution $Q_{W|S_N}$. Given a loss function $\ell : \mathcal{W} \times \mathcal{Z} \to \mathbb{R}^+$, where $\ell(w, Z)$ measures the "unfitness" or "error" of any $Z \in \mathcal{Z}$ with respect to a hypothesis $w \in \mathcal{W}$. We take $\ell$ as a continuous function and assume that $\ell$ is differentiable almost everywhere with respect to $w$. The goal of learning is to find a hypothesis $w$ that minimizes the population risk, and for any $w \in \mathcal{W}$, the population risk is defined as $L_\mu(w) \triangleq \mathbb{E}_{Z \sim \mu}[\ell(w, Z)]$. However, since only can partially observe $\mu$ via the sample $S_N$, we instead turn to use the empirical risk, defined as $L_{S_N}(w) \triangleq \frac{1}{N} \sum_{i=1}^N \ell(w, Z_i)$. Then the expected generalization error of $\mathcal{A}$ is defined as

$$\widetilde{error}(\mathcal{A}) \triangleq \mathbb{E}_{W, S_N}[L_\mu(W) - L_{S_N}(W)],$$

where the expectation is taken over $(S_N, W) \sim \mu^N \otimes Q_{W|S_N}$.

Consider the following variations of fine-tuning algorithms: tuning both $\mathbf{W}_k$ and $\mathbf{W}_q \& \mathbf{W}_v$ matrices (as in classic attention mechanism in fine-tuning), tuning only $\mathbf{W}_q \& \mathbf{W}_v$:

**Definition 1** (Fine-tuning algorithms). *Recalling **A unified framework for parameter-efficient fine-tuning**, we can model the fine-tuning process of the attention mechanism as $\mathbf{h} + \Delta \mathbf{h} = \mathbf{xW} + \mathbf{x}\Delta\mathbf{W}$. Let $\mathbf{W} = \{\mathbf{W}_i\}_{i=1}^L$ be a set of abstract parameter matrices related to a pretrained model, where each $\mathbf{W}_i$ is associated with the parameters $\mathbf{W}_q^i, \mathbf{W}_k^i, \mathbf{W}_v^i$. The indices $1, ..., L$ represent the layers of the model where these parameters are to be fine-tuned. Let $\mathcal{I} \subseteq \{1, ..., L\}$ denote the subset of layers selected for fine-tuning. Given a fine-tuning training set $S_N$, let $r$ denote the chosen lora-rank, and assume each tuned parameter is quantized to $q$ bits. Define the following algorithmic frameworks for selecting an adaptation $\Delta \mathbf{W} = \{\Delta \mathbf{W}_i\}_{i=1}^L$ (with other details left open to choice).*
*(1) $\mathcal{A}_{QKV}$: For each $i \in \mathcal{I}$, optimize $\{\mathbf{W}_q^i, \mathbf{W}_k^i, \mathbf{W}_v^i\}_{i \in \mathcal{I}}$ to fit the data $S_N$.*
*(2) $\mathcal{A}_{QV}$: For each $i \in \mathcal{I}$, optimize $\{\mathbf{W}_q^i, \mathbf{W}_v^i\}_{i \in \mathcal{I}}$ to fit the data $S_N$.*

Then we have the following theorem to bound the generalization error using the information-theoretic generalization framework.

**Theorem 1** (Generalization bounds on adapting $\mathbf{W}_q \& \mathbf{W}_v$ and/or $\mathbf{W}_k$). *Consider the algorithms of **Definition 1**. Assume the loss $\ell(\mathbf{W}, Z)$ is R-subGaussian under $(\Delta \mathbf{W}, Z) \sim P_{\Delta \mathbf{W}|\mathbf{W}} \times \mu$. Then,*

$$\widetilde{error}(\mathcal{A}_{QV}) \leq \sqrt{\frac{4R^2}{N} qr \sum_{i \in \mathcal{I}}(d_{in} + d_{out})},$$

$$\widetilde{error}(\mathcal{A}_{QKV}) \leq \sqrt{\frac{6R^2}{N} qr \sum_{i \in \mathcal{I}}(d_{in} + d_{out})},$$

*where $\mathbf{W}_q^i, \mathbf{W}_k^i, \mathbf{W}_v^i \in \mathbb{R}^{d_{in} \times d_{out}}$. See Appendix B.2 for a proof.*

**Remark 2** (Discussion of the advantages). *We can evaluate the empirical risk ($L_{S_N}$) by observing the model's performance on the dataset we have. If the generalization error (**Theorem 1**) is determined, it is at least possible to estimate the population risk ($L_\mu$). This generalization bound increases with the number of parameters being tuned, which grows as a function of $r$ and the dimensions of the parameter matrices. In Table 1, we know that with the same $r$ value, fine-tuning $\mathbf{W}_q \& \mathbf{W}_v$ consistently achieves results comparable to or even surpassing those of fine-tuning $\mathbf{W}_q, \mathbf{W}_k, \mathbf{W}_v$. This reduces the number of parameters for the same $r$, while **improving generalization bounds and potentially providing memory benefits**.*

## 4 CONVERGENCE ANALYSIS IN OPTIMIZATION

In Section 3, we have already demonstrated the generalization performance of the attention mechanism during fine-tuning. Our focus will now shift toward optimizing convergence efficiency. Some optimization observations have also been reported in previous works [20, 25, 15], such as: Li et al. [25] provide theoretical analyses of learning dynamics in transformers and observes a roughly two-stage process of self-attention. Meanwhile, He et al. [15] empirically show that the attention mechanism, particularly the value vector, stores the largest amount of memories and has the greatest influence during fine-tuning. However, there is not yet a satisfactory explanation for why this phenomenon occurs or how it can be effectively leveraged. In this section, we will explore these questions in more depth.

### 4.1 AN INSIGHT INTO INEFFICIENT LEARNING FOR VALUE MATRIX

We first discuss the optimization process of attention mechanism in the following simple case.

**Case 1.** *Omitting the scale factor for qualitative analysis in Eq.(1), we obtain:*

$$Attn(\mathbf{x}\mathbf{W}_q, \mathbf{C}\mathbf{W}_k, \mathbf{C}\mathbf{W}_v) = softmax\left(\mathbf{x}\mathbf{W}_q\mathbf{W}_k^T\mathbf{C}^T\right)\mathbf{C}\mathbf{W}_v.$$

*Intuitively, if $\mathbf{W}_q, \mathbf{W}_k, \mathbf{W}_v$ are initialized as random matrices close to zero and trained simultaneously, then in the initial step, $\nabla_{\mathbf{W}_k} L(\nabla_{\mathbf{W}_q} L)$ contains the term $\mathbf{W}_q(\mathbf{W}_k)$, which is close to 0. By contrast, $\nabla_{\mathbf{W}_v} L$ contains the softmax-normalized attention weights. Therefore, during the initial steps (in training), $\mathbf{W}_v$ intuitively grows at a much faster rate than $\mathbf{W}_k(\mathbf{W}_q)$.*

The work of [25] empirically exhibits **Case 1** with an approximately two-stage phenomenon: (1) In stage 1 (initial steps), the norms of $\mathbf{W}_k$ and $\mathbf{W}_q$ remain close to zero across all layers, while the norm of $\mathbf{W}_v$ increases significantly, accompanied by rapid changes in its orientation. (2) In stage 2, the norms of $\mathbf{W}_k$ and $\mathbf{W}_q$ begin to grow significantly, though much later than the $\mathbf{W}_v$ matrices. Briefly, in this case, $\mathbf{W}_v$ reaches a certain level of learning during training before $\mathbf{W}_k$ and $\mathbf{W}_q$ begin to learn. This suggests that when fine-tuning the model for downstream tasks, there may also be instances of inefficient learning in $\mathbf{W}_v$. Additionally, is there a fine-tuning strategy that could facilitate more effective learning for downstream tasks? **For instance, accelerating the learning of $\mathbf{W}_v$ in the early stages could potentially induce earlier learning in $\mathbf{W}_k$ and $\mathbf{W}_q$.**

Next, we present the second interesting phenomenon *Attention Matrices with Customized Learning Rate Leads to Better Convergence*. We use the General Language Understanding Evaluation (GLUE, [44]) to evaluate the fine-tuning performance of different fine-tuning strategies, which consists of several language tasks that evaluate the understanding capabilities of language models. Using LoRA, we fine-tune Roberta-base from the RoBERTa family [29] and Llama3.1-8b [2] on MNLI, QQP, QNLI, and SST2 tasks with varying learning rates ($\eta_{QK}, \eta_V$) to identify the optimal combination. Other empirical details are provided in Appendix C.1 and we evaluate the LLaMA3.1-8B model on more complex benchmarks in Appendix C.2.5. We present our empirical results using LoRA to fine-tune language models, as visualized in the heatmaps (Figure 1 and Figure 2).

In Figure 1 and Figure 2, we observe that (1) test accuracy consistently reaches its maximum for certain sets of learning rates where $\eta_{QK} < \eta_V$, outperforming the standard practice of setting $\eta_{QK}$ and $\eta_V$ equal. (2) More interestingly, the gap between the optimal choice of learning rates overall and the optimal choice when $\eta_{QK} = \eta_V$ varies across different tasks. This is probably due to the fact that harder task (like MNLI) requires more efficient feature learning. Additionally, we compare two optimal learning rate ($\eta_{QK}, \eta_V$) settings in Figure 2 (Left), the $\eta_V >> \eta_{QK}$ setting has a better convergence than $\eta_V = \eta_{QK}$ setting in Figure 2 (Right).

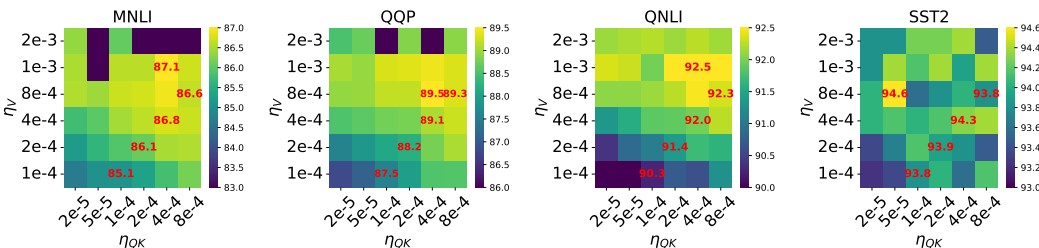

Figure 1: The test accuracy of RoBERTa-base fine-tuning was evaluated over 3 epochs for MNLI, QQP, and QNLI, and 6 epochs for SST-2, with a sequence length $T = 128$ and using half-precision (FP16). The LoRA hyperparameters were set to $\alpha = r = 8$. All reported values represent the average results across 3 random seeds. We highlight (1) the best overall accuracy and (2) the values where $\eta_V / \eta_{QK} = 1$. These values are shown in red. For better visualization, when accuracy is lower than a fixed threshold, we set it to threshold.

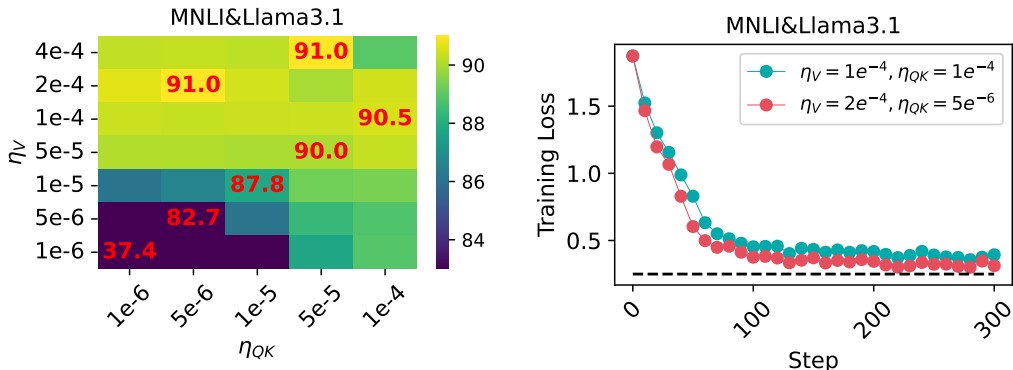

Figure 2: Left: The test accuracy of Llama3.1-8b fine-tuning was evaluated over 800 steps for MNLI. Key values like Figure 1 are also shown in red. Right: The training loss over 800 steps for MNLI fine-tuning on Llama3.1-8b, showing comparison between two optimal learning rate $(\eta_{QK}, \eta_V)$ settings in Left: (1) with $\eta_V = \eta_{QK}$ (2) with $\eta_V >> \eta_{QK}$.

It is also important to note that due to limited computational resources in our experiments, we use a sequence length of $T = 128$ and fine-tune for only 3 epochs on MNLI and QQP. Therefore, it is expected that our test accuracies may be lower than those reported by Hu et al. [20], where the authors fine-tune RoBERTa-base with a sequence length of $T = 512$ (for MNLI) and for more epochs (30 for MNLI). We do not include confidence intervals for clearer visualization, however, the fluctuations remain within acceptable limits. See Figure 2 (Right) for instance. In Appendix C.2, we provide additional results including the training loss.

### 4.2 CONVERGENCE ANALYSIS FOR LEARNING RATE

It naturally raises the question of why $\eta_{QK}$ and $\eta_V$ should be set differently. In practice, most state-of-the-art models have a large width (embedding dimension), making it worthwhile to examine the training dynamics as the width approaches infinity.

**Starting with a Toy setting.** Revisiting **Definition 1**, we have $\Delta \mathbf{h} = \text{softmax}(\mathbf{x}\mathbf{A})\mathbf{B}$. In the case of a linear attention mechanism, we instead have $\Delta \mathbf{h} = \mathbf{x}\mathbf{A}\mathbf{B}$. Then consider the following toy setting

$$f(x) = x(W^* + a^T b),$$

where $W^* \in \mathbb{R}^{n \times 1}$ are the fixed[4] pre-trained weights, $b \in \mathbb{R}, a \in \mathbb{R}^{1 \times n}$ are adaptation weights, $x \in \mathbb{R}^n$ is the model input (This corresponds to $r = 1$ in **Definition 1**). The training goal is to

---

[4]Here, we primarily focus on the case of $\Delta \mathbf{W}$ to provide insightful theoretical results.

minimize the loss $\mathcal{L}(\theta) = \frac{1}{2}(f(x) - y)^2$ where $\theta = (a, b)$ and $(x, y)$ is an input-output datapoint[5]. Similar to LoRA, we generally aim to initialize the product $a^T b$ to zero, ensuring that fine-tuning starts from the pre-trained model. This requires at least one of the weights, $a$ (related to $\mathbf{W}_q \& \mathbf{W}_k$) or $b$ (related to $\mathbf{W}_v$), to be initialized to zero. If both are initialized to zero, $\mathbf{W}_q \& \mathbf{W}_k$ learning cannot occur efficiently in init steps, as discussed in Section 4.1 (More detailed initialization settings are shown in Appendix B.3).

And we assume that $x = \Theta(1)$, meaning that the input coordinates remain of the same order as the width increases. In the subsequent analysis, we examine how the fine-tuning dynamics evolve as the model width $n$ increases.

To streamline the analysis, we assume $W^* = 0$, a common simplification that can be applied without loss of generality. This assumption is implemented by setting $\hat{y} = y - xW^*$. We denote the fine-tuning step by using subscript $t$. Let $U_t = f_t(x) - y$, the gradients are then computed as:

$$\frac{\partial \mathcal{L}}{\partial a_t} = xU_t b_t, \quad \frac{\partial \mathcal{L}}{\partial b_t} = xa_t^T U_t.$$

And at step $t$ with learning rate $\eta_a, \eta_b > 0$, we have

$$\Delta f_t \triangleq f_t(x) - f_{t-1}(x)$$
$$= -\underbrace{\eta_a ||x||^2 U_{t-1} b_{t-1}^2}_{\delta_t^1} - \underbrace{\eta_b (xa_{t-1}^T)^2 U_{t-1}}_{\delta_t^2} + \underbrace{\eta_a \eta_b ||x||^2 (xa_{t-1}^T) U_{t-1}^2 b_{t-1}}_{\delta_t^3}.$$

**Remark 3.** *The output update is influenced by three key terms. The first two items $\delta_t^1, \delta_t^2$ (order one in $\eta_a/\eta_b$) represent linear contributions to the update, meaning they result from changes in the model output when either $a$ is updated with $b$ held constant, or vice versa. The last item $\delta_t^3$ (order two in $\eta_a\eta_b$) corresponds to a multiplicative update that captures the combined effects of changes in both $a$ and $b$. As we scale the width[6], **the desirable feature updates are such that $\Delta f_t = \Theta(1)$,** ensuring they remain unaffected by this scaling (the updates do not explode with width, see $x$ for more details). Ideally, we aim for both $\delta_t^1$ and $\delta_t^2$ to be $\Theta(1)$. If this condition isn't met, it indicates that either $a$ or $b$ is not being updated efficiently. For example, if $\delta_t^1 = o(1)$, it suggests that as $n \to \infty$, the model behaves as if $a$ is essentially fixed, with only $b$ being trained. We say that the **feature learning in the attention mechanism is efficient** when $\delta_t^i = \Theta(1)$ for $i \in \{1, 2\}$ and all $t > 1$, it means that both $a$ and $b$ parameter updates significantly contribute to the change in $f_t(x)$. We will see that when both $\delta_t^1$ and $\delta_t^2$ are $\Theta(1)$, the term $\delta_t^3$ is also $\Theta(1)$.*

Let us assume that we train the model with gradient descent with learning rate $\eta_a = \Theta(n^{c_a}), \eta_b = \Theta(n^{c_b})$ for some $c_a, c_b \in \mathbb{R}$. In the study by Yang et al. [49], it is noted that the training dynamics primarily involve operations such as matrix-vector products and the summation of vectors or scalars. Given the nature of these operations, it is easy to see that any quantity in the training dynamics should be of order $n^\gamma$ for some $\gamma \in \mathbb{R}$. We write $v = \Theta(n^{\gamma[v]})$, for any quantity $v$ in the training dynamics. When $v$ is a vector, we use the same notation when all entries of $v$ are $\Theta(n^{\gamma[v]})$ (See Appendix B.4 for the formal definition of $\gamma$).

With reference to the method of Hayou et al. [13], we start from the initialization in **Starting with a Toy setting**, we have $f_0(x) = 0$. Feature learning of attention mechanism is efficient when $\delta_t^i = \Theta(1)$ for $i \in \{1, 2\}$ and all $t > 1$, and $f_t(x) = \Theta(1)$ for $t > 1$. This can be interpreted as:

$$\begin{cases} c_a + 1 + 2\gamma[b_{t-1}] = 0 & (\delta_t^1 = \Theta(1)) \\ c_b + 2\gamma[xa_{t-1}^\top] = 0 & (\delta_t^2 = \Theta(1)) \\ \gamma[xa_{t-1}^\top] + \gamma[b_{t-1}] = 0 & (f_{t-1}(x) = \Theta(1)), \end{cases}$$

which, after simple calculations, implies that $c_a + c_b = -1$. Notice that the above also leads to the $c_a + c_b + 1 + \gamma[xa_{t-1}^\top] + \gamma[b_{t-1}] = 0$ $(\delta_t^3 = \Theta(1))$. This is only a necessary condition. In the following section, we will provide theoretical conclusions in the toy model setting that offer guidance for real-world experiments.

---

[5]To simplify the analysis, we assume that the fine-tuning dataset consists of a single sample, though our analysis can be easily generalized to multiple samples. All conclusions remain essentially valid when $(a, b)$ are matrices.

[6]This property is generally satisfied in practice when the model width is large (e.g., $n \approx 800$ for Roberta-base and $n \approx 4000$ for Llama3.1-8b).

**Theorem 2** (Efficient fine-tuning in attention mechanism (Informal)). *In the case of **Starting with a Toy setting**, with $\eta_a = \Theta(n^{-1})$ and $\eta_b = \Theta(1)$, we have for all $t > 1$, $i \in \{1, 2, 3\}, \delta_t^i = \Theta(1)$. In other words, the feature learning of attention mechanism is efficient when $\eta_{QK}(\eta_a) = \Theta(n^{-1}), \eta_V(\eta_b) = \Theta(1)$. We denote $\eta_V/\eta_{QK}$ as $\lambda$. We refer the reader to Appendix B.5 for more details on the proof.*

**Remark 4.** *In practice, **Theorem 2** implies that the learning rate for $\mathbf{W}_v$ should be generally much larger than that of $\mathbf{W}_q \& \mathbf{W}_k$ in fine-tuning. We verify that this scaling is valid for general neural network models in Section 4.1. Naturally, the optimal ratio $\lambda$ depends on the architecture and the fine-tuning task through the constants in 'Θ'. This represents a limitation of the asymptotic results, as they do not provide insights into how the task and neural architecture influence these constants. We will further address this issue in our future work.*

**A summary of the main theoretical analyses.** According to the traditional statistical learning viewpoint, performance can be defined by the sum of optimization error and generalization error. Our theoretical analyses in Sections 3 and 4 correspond to generalization and optimization, respectively. In Section 3 (generalization, storage-friendly), we give **Theorem 1** (Information-theoretic genralization bounds), showing that with the same r value, fine-tuning $\mathbf{W}_q \& \mathbf{W}_v$ consistently achieves results comparable to or even surpassing those of fine-tuning $\mathbf{W}_q, \mathbf{W}_k, \mathbf{W}_v$. This reduces the number of parameters for the same $r$, while improving generalization bounds and potentially providing memory benefits. In Section 4 (optimization, time-friendly), we discuss the learning dynamics in fine-tuning attention mechanism, and we illustrate (**Theorem 2**) that the feature learning of attention mechanism is efficient when the learning rate for $\mathbf{W}_v$ should be generally much larger than that of $\mathbf{W}_q \& \mathbf{W}_k$ in fine-tuning. Building on our experimental and theoretical insights, one can develop new algorithms to improve the effectiveness (e.g., storage, and time) of fine-tuning (Example in Section 5).

## 5 AN EXAMPLE OF IMPROVING FINE-TUNING METHODS

Based on all our exciting insights, it becomes intuitive to design lightweight attention-based fine-tuning improvements, particularly for downstream tasks. To illustrate how theoretical analysis effectively guides experimental procedures, we propose an example method where we freeze the $\mathbf{W}_k$ and fine-tuning the $\mathbf{W}_q \& \mathbf{W}_v$ using different learning rates. This procedure is reported in Figure 5.

**How to set the ratio $\lambda$?** Naturally, as discussed in **Remark 4**, the optimal ratio $\lambda$ depends on the architecture and the fine-tuning task via the constants in $\Theta$ in **Theorem 2**. This is a limitation of these asymptotic results since they do not offer any insights on how the constants are affected by the task and the neural architecture. However, we can still employ some heuristic methods, such as: we can select an appropriate range by conducting a certain amount of experiments, as shown in Figure 1, it seems that a ratio of order $2^1 - 2^4$ is optimal. Moreover, $\lambda$ should not be too large; otherwise, as shown in the MNLI subplot in Figure 1, the model's performance will collapse.

**Experimental setup.** We conduct experiments on widely adopted benchmark datasets [44] and Roberta-base model [29]. We selected two mainstream baselines: Full Fine-tuning, LoRA [20] and DoRA [28]. Additionally, we adapt **only the attention weights** for downstream tasks, keeping the other modules frozen to maintain simplicity and validate the theoretical guidance through experiments. In our experiments, we evaluated the performance for $\lambda$ values of 2, 4, and 8 (one can also determine a general optimal ratio through experiments, and even apply different settings across different layers of the model). We report the average results based on 3 random seeds, as shown in Table 2. The hyperparameter settings for the experiments can be found in Appendix C.1.2 and the base model performance for each task can be seen in Table 2 and Appendix C.2.2. We also have added ablation experiments on different models (Mistral-7B [3]) in Appendix C.2.4.

**Results.** We leverage our theoretical results (**Theorem 1** and **Theorem 2**) to enhance the efficiency of existing fine-tuning methods, such as Full Fine-tune and LoRA, on downstream tasks. As shown in Table 2, the improved fine-tuning approach not only outperforms the original version but also significantly reduces the number of parameters. For instance, on the MRPC task, *LoRA (QV) $r = 16, \lambda = 8$ (1.77M)* achieves better performance compared to *Full Fine-tune (QKV) (21.85M)* and *LoRA (QKV) $r = 16$ (2.07M)*. This series of experiments clearly demonstrates that our theoretical insights effectively enhance fine-tuning algorithms, particularly in terms of memory usage

Table 2: Comparison of fine-tuning methods across GLUE benchmark. We report results on development set, Pearson correlation for STS-B, Matthew's correlation for CoLA, average accuracy for MNLI (matched and mismatched), and accuracy for other tasks. The best results on each dataset are shown in **bold** and the second best results are shown in underline. The QKV(QV) setting refers to fine-tuning $\mathbf{W}_q, \mathbf{W}_k, \mathbf{W}_v (\mathbf{W}_q, \mathbf{W}_v)$. It is noted that the total number of parameters in the Roberta-base model is 124.65M. $\lambda$ means $\eta_V = \lambda\eta_Q$ and $r$ is the LoRA rank, and a larger $\lambda$ does not necessarily lead to better performance.

| Method | Trainable #Param (M) | RTE | STS-B | MRPC | CoLA | MNLI | SST-2 | QQP | QNLI |
|---|---|---|---|---|---|---|---|---|---|
| Before Fine-tune | 0 | 45.12 | -3.18 | 66.66 | 1.09 | 32.95 | 49.31 | 44.72 | 50.81 |
| Full Fine-tune (QKV) | 21.85 | 73.64 | 90.49 | 84.55 | 60.34 | 86.68 | 93.23 | 90.48 | 92.37 |
| LoRA (QKV) $r = 8$ | 1.62 | 70.76 | 90.25 | 85.04 | 58.03 | 86.70 | 93.92 | 89.15 | 92.17 |
| LoRA (QKV) $r = 16$ | 2.07 | 70.39 | 90.25 | 86.03 | 58.04 | 86.78 | 93.92 | 89.26 | 92.18 |
| DoRA (QKV) $r = 8$ | 1.06 | 70.75 | 90.39 | 85.78 | 56.79 | 86.73 | 93.58 | 89.34 | 92.22 |
| DoRA (QKV) $r = 16$ | 1.51 | 70.40 | 90.31 | 86.03 | 57.81 | 86.77 | 93.92 | 89.30 | 92.48 |
| Full Fine-tune (QV) $\lambda = 2$ | 14.76 | 73.53 | 91.01 | 86.02 | 60.57 | 62.03 | 93.11 | **90.56** | 91.96 |
| Full Fine-tune (QV) $\lambda = 4$ | 14.76 | 72.29 | 90.56 | 87.01 | **61.88** | 35.44 | 91.05 | 89.81 | 88.85 |
| Full Fine-tune (QV) $\lambda = 8$ | 14.76 | 72.29 | 90.02 | 88.97 | 61.86 | 35.44 | 84.75 | 85.93 | 50.54 |
| LoRA (QV) $r = 8, \lambda = 2$ | 1.48 | 71.84 | 90.37 | 86.02 | 58.54 | 86.85 | 94.03 | 89.47 | 92.33 |
| LoRA (QV) $r = 8, \lambda = 4$ | 1.48 | 75.09 | 90.83 | 87.01 | 59.56 | 86.95 | 94.04 | 90.09 | 92.86 |
| LoRA (QV) $r = 8, \lambda = 8$ | 1.48 | 76.13 | 90.75 | 88.97 | **61.88** | 86.93 | 93.46 | 90.01 | 92.34 |
| LoRA (QV) $r = 16, \lambda = 2$ | 1.77 | 70.39 | 90.46 | 86.03 | 58.55 | 86.83 | **94.38** | 89.77 | 92.33 |
| LoRA (QV) $r = 16, \lambda = 4$ | 1.77 | 76.17 | **91.05** | 87.99 | 60.06 | 87.19 | 94.03 | 90.30 | 92.73 |
| LoRA (QV) $r = 16, \lambda = 8$ | 1.77 | 72.92 | 90.96 | **89.95** | 59.31 | **87.31** | 93.92 | 90.43 | 92.95 |
| DoRA (QV) $r = 8, \lambda = 2$ | 0.90 | 71.12 | 90.29 | 87.01 | 58.54 | 87.08 | 93.96 | 89.60 | 92.60 |
| DoRA (QV) $r = 8, \lambda = 4$ | 0.90 | 75.45 | 90.82 | 86.76 | 60.32 | 86.98 | 93.81 | 90.33 | 92.97 |
| DoRA (QV) $r = 8, \lambda = 8$ | 0.90 | 70.76 | 90.38 | 87.75 | 57.01 | 87.12 | 94.15 | 90.45 | 92.48 |
| DoRA (QV) $r = 16, \lambda = 2$ | 1.20 | 69.68 | 90.53 | 87.75 | 59.31 | 87.09 | 93.92 | 89.68 | 92.70 |
| DoRA (QV) $r = 16, \lambda = 4$ | 1.20 | 76.16 | 90.77 | 88.48 | 60.84 | 86.96 | 94.15 | 90.34 | **93.01** |
| DoRA (QV) $r = 16, \lambda = 8$ | 1.20 | **77.26** | 90.83 | 88.96 | 60.32 | 87.10 | 94.17 | 90.46 | 92.80 |

and optimization efficiency. Moreover, these theoretical results can guide the improvement of other fine-tuning algorithms and even aid in the design of more efficient ones.

## 6    CONCLUSION AND LIMITATION

In this paper, we present our key findings in fine-tuning attention mechanism: *Unequal Importance of Attention Matrices*—optimizing the $\mathbf{W}_v$ matrix significantly improves performance compared to the $\mathbf{W}_k$ matrix. Fine-tuning only the $\mathbf{W}_q$ and $\mathbf{W}_v$ matrices is computationally efficient and can yield results that match or surpass fine-tuning all three matrices $\mathbf{W}_q$, $\mathbf{W}_k$, and $\mathbf{W}_v$. *Attention Matrices with Customized Learning Rate Leads to Better Convergence*—using distinct learning rates for these matrices is essential for optimal performance, with a higher learning rate for $\mathbf{W}_v$ speeding up convergence. While theoretical analysis of these phenomena is limited, this paper provides insights from two angles: *Generalization*—fine-tuning only $\mathbf{W}_q$ and $\mathbf{W}_v$ improves generalization and memory efficiency, and *Optimization*—using different learning rates enhances the efficiency of feature learning in the attention mechanism, leading to more effective fine-tuning. Our analysis provides a theoretical foundation for the configuration and improvement of lightweight algorithms in LLMs fine-tuning. However, further studies are required on (i) how the task and neural architecture influence the optimal ratio $\lambda$ and (ii) whether the results about attention hold true for tasks beyond natural language processing. These studies will further deepen our understanding of attention-based fine-tuning in LLMs.

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

# A    MORE RELATED WORKS

**Attention mechanism analysis.** A key component of transformers is the attention mechanism, which dates back to [9]. Initially designed to capture long-range signals in sequential inputs by mixing individual tokens, it has also been utilized to capture general structures in input data. After the fully-attention-based language model has appeared [43, 5], the research community gets interested in the functionality and benefits of the attention. For instance, transformers implicitly favor hierarchical interpretations of input sequences [21], the computational graphs tend to be tree-structured [30, 32]. Theoretical analysis of training dynamics sheds light on how to identify key tokens [41], select a few relevant tokens only (which is called localized attention) or select many tokens uniformly [4], and learn topic structure [25]. Besides, considerable works [36, 1, 51] try to understand in-context learning capabilities from the perspective of gradient descent with attention.

**Scaling for neural networks.** Scaling refers to the process of enlarging a specific ingredient of a model to enhance its overall performance [18]. The method is straightforward: extend the width or depth of a neural network towards infinity, analyze how this limit is influenced by hyperparameters like the learning rate and initialization variance during training, and then establish well-founded choices for these hyperparameters to achieve a specific objective [17, 38, 11, 47, 49, 14, 10, 50, 13]. In the theory of scaling of neural networks, one usually tracks the asymptotic behaviour of key quantities as we scale some model ingredient, it is a standard approach used to derive scaling rules for initialization [38], activation function [11], network parametrization [50]. In this paper, we are interested in scaling model capacity via the width n for the fact that most state-of-the-art pre-trained models have large width. Examples of the infinite-width limit can be found in studies focused on initialization methods [17, 47], or more comprehensive approaches to network parameterization. For instance, Yang et al. [49] introduced μP, a parameterization technique for neural networks that guarantees feature learning in the infinite-width limit, providing specific scaling rules for both architecture and learning rates to optimize feature learning [48, 49].

**Parameter-efficient fine-tuning.** Fine-tuning all the parameters of a large language models, known as full fine-tuning, is highly computationally expensive. To reduce the computational cost, various parameter-efficient fine-tuning (PEFT) methods have been proposed [7], which only fine-tune a small number of (extra) model parameters. PEFT methods can be divided into two categories from the perspective of whether extra parameters are involved: (1) extra-parameter methods, freeze all of the original parameters of an LLM and insert a set of learnable parameters to optimize the model input or model layers suach as adapter tuning [19], prompt tuning [23] and prefix tuning [24]; (2) intra-parameter methods freeze most of the original parameters of an LLM and only tune a small number of parameters of the LLM such as LoRA [20]. Furthermore, He et al. [16] present a unified framework that establishes connections between PEFT methods and Zhu et al. [53] formally identify and investigate asymmetry in the roles of low-rank adapter matrices in LoRA fine-tuning.

# B    OMITTED PROOFS AND ADDITIONAL RESULTS

## B.1    THE CONNECTION BETWEEN PREFIX TUNING AND LORA.

Here, we provide an alternative view of Prefix tuning (without loss of generality, we ignore the softmax scaling factor for ease of notation):

$$\text{Attn}(\mathbf{x}\mathbf{W}_q, \text{concat}(\mathbf{P}_k, \mathbf{C}\mathbf{W}_k), \text{concat}(\mathbf{P}_v, \mathbf{C}\mathbf{W}_v)$$

$$= \text{softmax}(\mathbf{x}\mathbf{W}_q\text{concat}(\mathbf{P}_k, \mathbf{C}\mathbf{W}_k)^T)\begin{pmatrix} \mathbf{P}_v \\ \mathbf{C}\mathbf{W}_v \end{pmatrix}$$

$$= (1 - \lambda(\mathbf{x}))\text{softmax}(\mathbf{x}\mathbf{W}_q\mathbf{W}_k^T\mathbf{C}^T)\mathbf{C}\mathbf{W}_v + \lambda(\mathbf{x})\text{softmax}(\mathbf{x}\mathbf{W}_q\mathbf{P}_k^T)\mathbf{P}_v$$

$$= (1 - \lambda(\mathbf{x}))\overbrace{\text{Attn}(\mathbf{x}\mathbf{W}_q, \mathbf{C}\mathbf{W}_k, \mathbf{C}\mathbf{W}_v)}^{\text{standard attention}} + \lambda(\mathbf{x})\overbrace{\text{Attn}(\mathbf{x}\mathbf{W}_q, \mathbf{P}_k, \mathbf{P}_v)}^{\text{independent of C}},$$

where $\lambda(\mathbf{x}) = \frac{\sum_i \exp(\mathbf{x}\mathbf{W}_q\mathbf{P}_k^T)_i}{\sum_i \exp(\mathbf{x}\mathbf{W}_q\mathbf{P}_k^T)_i + \sum_j \exp(\mathbf{x}\mathbf{W}_q\mathbf{W}_k^T\mathbf{C}^T)_j}$ is a scalar that represents the sum of normalized attention weights on the prefixes. Notice that the first term in blue represents the original attention mechanism without prefixes, while the second term in green introduces a position-wise ad-

justment independent of $\mathbf{C}$. It provides an alternative perspective on Prefix tuning, where a position-wise modification is applied to the original attention output $\mathbf{h}$ via linear interpolation:

$$\mathbf{h} \leftarrow (1 - \lambda(\mathbf{x})\mathbf{h} + \lambda(\mathbf{x})\Delta\mathbf{h}, \quad \Delta\mathbf{h} := \text{softmax}(\mathbf{x}\mathbf{W}_q\mathbf{P}_k^T)\mathbf{P}_v \triangleq \text{softmax}(\mathbf{x}\mathbf{A})\mathbf{B}.$$

### B.2 PROOF OF THEOREM 1

The origin form of the mutual information based bound is predicated on a sample-specific MI, which quantifies the shared information between the output variable $\mathbf{W}$ and the input sample set $S_N$. The following lemma shows the result:

**Lemma 1.** *(Xu and Raginsky [46, Theorem 1.]). Assume the loss $\ell(\mathbf{W}, Z)$ is R-subGaussian for any $\mathbf{W} \in \mathcal{W}$, then*

$$\widetilde{error}(\mathcal{A}) \leq \sqrt{\frac{2R^2}{N} I(\mathbf{W}; S_N)},$$

*where $I(\mathbf{W}; S_N) = D_{KL}(Q_{\mathbf{W},S_N} \| Q_{\mathbf{W}} \otimes Q_{S_N})$ is the mutual information and $D_{KL}$ denotes the KL divergence.*

Unroll the terminal parameters' mutual information $I(\mathbf{W}; S_N)$ to the full trajectories' mutual information will get:

**Lemma 2.** *Let **Definition 1** hold, then $I(\mathbf{W} + \Delta\mathbf{W}; S_N|\mathcal{A}) \leq I(\Delta\mathbf{W}; S_N|\mathcal{A}, \mathbf{W}).$*

*Proof.*

$$\begin{aligned}
&I(\mathbf{W} + \Delta\mathbf{W}; S_N|\mathcal{A}) \\
&\leq I(\mathbf{W}, \Delta\mathbf{W}; S_N|\mathcal{A}) && (*) \\
&= I(\mathbf{W}; S_N|\mathcal{A}) + I(\Delta\mathbf{W}; S_N|\mathcal{A}, \mathbf{W}) && (**) \\
&= I(\Delta\mathbf{W}; S_N|\mathcal{A}, \mathbf{W}).
\end{aligned}$$

where Eq. (*) is by the data processing inequality (e.g., $Z - (X, Y) - (X + Y)$ form a Markov chain then $I(X + Y, Z) \leq I(X, Y; Z)$), Eq. (**) is by the chain rule of the mutual information, and $I(\mathbf{W}; S_N) = 0$ for $\mathbf{W}$ is independent of $S_N$. □

Then combine **Lemma 1** and **Lemma 2**, we can get: $\widetilde{error}(\mathcal{A}) \leq \sqrt{\frac{2R^2}{N} I(\Delta\mathbf{W}; S_N|\mathcal{A}, \mathbf{W})}$.
We consider the case of tuning $\mathbf{W}_q \& \mathbf{W}_v$ only first. Applying the above results, note that here

$$I(\Delta\mathbf{W}; S_N|\mathcal{A}_{QV}, \mathbf{W}) = I(\{\mathbf{W}_q^i, \mathbf{W}_v^i\}_{i \in \mathcal{I}}; S_N|\mathcal{A}_{QV}, \mathbf{W}),$$

where we have used the data processing inequality (DPI), noting that the $\mathbf{W}_k^i$ are here considered fixed constant matrices as they are not trained.

We can now bound this expression as

$$I(\{\mathbf{W}_q^i, \mathbf{W}_v^i\}_{i \in \mathcal{I}}; S_N|\mathcal{A}_{QV}, \mathbf{W}) \leq H(\{\mathbf{W}_q^i, \mathbf{W}_v^i\}_{i \in \mathcal{I}}) \leq 2qr \sum_{i \in \mathcal{I}}(d^i + k^i),$$

where $\mathbf{W}_q^i, \mathbf{W}_k^i, \mathbf{W}_v^i \in \mathbb{R}^{d_{in} \times d_{out}}$, and we have noted that mutual information is upper bounded by discrete entropy, and entropy in turn is upper bounded by the uniform distribution over its possible support set (q bits in each of $r \sum_{i \in \mathcal{I}}(d_{in} + d_{out})$ dimensions). The bounds for the other algorithms are similar.

### B.3 INITIALIZATION DISCUSSION

Following standard initialization schemes (e.g., LeCun Init and He Init [22, 17]), one generally consider a Gaussian initialization of the weights as follows: $a_i \sim \mathcal{N}(0, \sigma_a^2), \quad b \sim \mathcal{N}(0, \sigma_b^2)$ (The Gaussian distribution can be substituted with any other distribution that has finite variance). Revisiting **Starting with a Toy setting**, $a \in \mathbb{R}^{1 \times n}, b \in \mathbb{R}$. Thus, one should set $\sigma_a^2 = \Theta(n^{-1}), \sigma_b^2 = 0$ to ensure $xa^T$ does not explode with width ($xa^T = \Theta(1)$), for a non-zero initialization for $a$. This is justified by the Central Limit Theorem (See [49] for more technical details). And if we choose a non-zero initialization for $b$, one should make sure that $\sigma_b^2 = \Theta(1), \sigma_a^2 = 0$. And we will consider these two initialization schemes to show our theoretical understanding.

### B.4 GAMMA FUNCTION

**Why introduce the Gamma function?**
In Section 4.2, the learning rate $\eta_a = \Theta(n^{c_a}), \eta_b = \Theta(n^{c_b})$ for some $c_a, c_b \in \mathbb{R}$. And in Appendix B.3 we assume that the init weights are also scale polynomially with n, it is evident that preactivations, gradients, and weight updates all exhibit asymptotic polynomial growth in n.
**Operations.**
We write $v = \Theta(\gamma[v])$ to capture it, and some elementary operations (Given two real-valued variables $v_1, v_2$):

- Multiplication. $\gamma[v_1 \times v_2] = \gamma[v_1] + \gamma[v_2]$.

- Addition. Generally, we have $\gamma[v_1 + v_2] = \max(\gamma[v_1], \gamma[v_2])$. The only instance where this does not hold is when $v_1 = -v_2$. This is typically a zero-probability event if the two variables are random variables that are not perfectly correlated, which is the case in most scenarios where we apply this formula (Appendix B.5).

### B.5 PROOF OF THEOREM 2

**Theorem 2.**[Efficient fine-tuning in attention mechanism (Informal)]
In the case of **Starting with a Toy setting**, with $\eta_a = \Theta(n^{-1})$ and $\eta_b = \Theta(1)$, we have for all $t > 1, i \in \{1, 2, 3\}, \delta_t^i = \Theta(1)$. In other words, the feature learning of attention mechanism is efficient when $\eta_{QK}(\eta_a) = \Theta(n^{-1}), \eta_V(\eta_b) = \Theta(1)$.

*Proof.* In Section 3, we say that the feature learning of attention mechanism is efficient when $\delta_t^i = \Theta(1)$ for all $t, i \in \{1, 2, 3\}$. Using the elementary formulas from Appendix B.4, we can get (for all $t$):

$$\begin{cases} \gamma[\eta_a] + 1 + 2\gamma[b_{t-1}] = 0 & \left(\delta_t^1 = \Theta(1)\right) \\ \gamma[\eta_b] + 2\gamma[xa_{t-1}^\top] = 0 & \left(\delta_t^2 = \Theta(1)\right) \\ \gamma[\eta_a] + \gamma[\eta_b] + 1 + \gamma[xa_{t-1}^\top] + \gamma[b_{t-1}] = 0 & \left(\delta_t^3 = \Theta(1)\right). \end{cases}$$

Simple calculations yield $\gamma[\eta_a] + \gamma[\eta_b] = -1$. Further consider the gradient update from $t-1$ to $t$, the recursive formulas are given by:

$$\begin{cases} \gamma[xa_t^\top] = \max\left(\gamma[xa_{t-1}^\top], \gamma[\eta_a] + 1 + \gamma[b_{t-1}]\right) \\ \gamma[b_t] = \max\left(\gamma[b_{t-1}], \gamma[\eta_b] + \gamma[xa_{t-1}^\top]\right) \end{cases}$$

Starting from $t = 1$. In both initialization schemes discussed in Appendix B.3, we have to set $\gamma[\eta_b] = 0$ and $\gamma[\eta_a] = -1$ to ensure that $\gamma[f_t] = \gamma[xa_t^T] + \gamma[b_t] = 0$:
(1) $\sigma_a^2 = \Theta(n^{-1}), \sigma_b^2 = 0$. We have $\gamma[xa_1^T] = \gamma[xa_0^T] = 0$, $\gamma[b_1] = \gamma[\eta_b(xa_0^T)y] = \gamma[\eta_b]$. Therefore, for $t = 2$, $\gamma[xa_2^T] = \max(0, \gamma[\eta_a] + 1 + \gamma[\eta_b]) = \max(0, 0) = 0$, $\gamma[b_2] = \max(\gamma[\eta_b], \gamma[\eta_b] + 0) = \gamma[\eta_b]$, this holds for $t \geq 1$ by induction.
(2) $\sigma_a^2 = 0, \sigma_b^2 = \Theta(1)$. We have $\gamma[b_1] = \gamma[b_0] = 0$, $\gamma[xa_1^T] = \gamma[\eta_a||x||^2 U_0 b_0^2] = \gamma[\eta_a] + 1$. Therefore, for $t = 2$, $\gamma[b_2] = \max(0, \gamma[\eta_b] + \gamma[\eta_a] + 1) = \max(0, 0), \gamma[xa_2^T] = \max(\gamma[\eta_a] + 1, \gamma[\eta_a] + 1 + 0) = \gamma[\eta_a] + 1$, this holds for $t \geq 1$ by induction.
To sum up, setting $\eta_{QK}(\eta_a) = \Theta(n^{-1}), \eta_V(\eta_b) = \Theta(1)$ ensures efficient fine-tuning in attention mechanism. □

## C EXTENSION TO EXPERIMENTS

### C.1 EMPIRICAL DETAILS

#### C.1.1 GLUE TASKS WITH ROBERTA

For our experiments with Roberta-base models, finetuned on GLUE tasks, we use the following setup:

**Tasks.** MNLI, QQP, SST2, QNLI

**Training Algorithm.** AdamW with $\beta_1 = 0.9, \beta_2 = 0.99, \epsilon = 1e - 8$, linear schedule, no warmup.

**Targert Modules for Fine-tuning.** 'query', 'key' and 'value'.

**Learning rate.**
(1) For Table 1, $\eta_{QK} = \eta_V = 5e^{-5}$.
(2) For Figure 1,
$\eta_{QK} = \{2e^{-5}, 5e^{-5}, 1e^{-4}, 2e^{-4}, 4e^{-4}, 8e^{-4}\},$
$\eta_V = \{1e^{-4}, 2e^{-4}, 4e^{-4}, 8e^{-4}, 1e^{-3}, 2e^{-3}\}$

**GPUs.** Nvidia A800.

**Other Hyperparameters.** Sequence length $T = 128$, train batch size $batchsize = 32$, number of train , number of random seeds $s = 3$.
(1) For Table 1, epochs E = 6 (E = 10 for SST2).
(2) For Figure 1, epochs E = 3 (E = 6 for SST2).

### C.1.2 TRAINING HYPERPARAMETERS

**Training hyperparameters.**

| Corpus | length | learning rate | batch size | epochs |
|--------|--------|---------------|------------|--------|
| RTE    | 128    | 1e-04         | 32         | 20     |
| MRPC   | 128    | 1e-04         | 32         | 20     |
| STS-B  | 128    | 1e-04         | 32         | 20     |
| CoLA   | 128    | 1e-04         | 32         | 20     |
| SST-2  | 128    | 1e-04         | 32         | 10     |
| QNLI   | 128    | 1e-04         | 32         | 10     |
| QQP    | 128    | 1e-04         | 32         | 10     |
| MNLI   | 128    | 1e-04         | 32         | 10     |

Table 3: Training hyperparameters for different datasets. More details can be seen in our code.

- For *Full Fine-tune (QKV)* and *LoRA (QKV)*, we use $\eta_Q = \eta_K = \eta_V = $ 1e-04.
- For the improved methods, we use $\eta_Q = $ 1e-04, $\eta_V = \lambda \times$ 1e-04.

The hyperparameter settings here differ from those in Table 1 and Figure 1, so the results may show slight variations.

### C.1.3 MNLI TASK WITH LLAMA3.1-8B

For our experiments with Llama3.1-8b models, finetuned on MNLI, we use the following setup:

**Training Algorithm.** AdamW with $\beta_1 = 0.9, \beta_2 = 0.999, \epsilon = 1e - 6$, constant schedule.

**Targert Modules for Fine-tuning.** 'q_proj, k_proj, v_proj'.

**Learning rate grid.**
For Table 1, $\eta_{QK} = \eta_V = 1e^{-5}$.
Else:
$\eta_{QK} = \{1e^{-6}, 5e^{-6}, 1e^{-5}, 5e^{-5}, 1e^{-4}\},$
$\eta_V = \{1e^{-6}, 5e^{-6}, 1e^{-5}, 5e^{-5}, 1e^{-4}, 2e^{-4}, 4e^{-4}\}$

**Hyperparameters.** LoRA rank $r = 16, \alpha = 16$, and dropout 0.1. Precision FP16. Sequence length $T = 128$, train batch size $batchsize = 128$.

**GPUs.** Nvidia A800.

**Before fine-tuning model performance.** QQP: 55.08 , MNLI: 33.34

## C.2 Empirical Results

### C.2.1 GLUE Tasks Train Loss

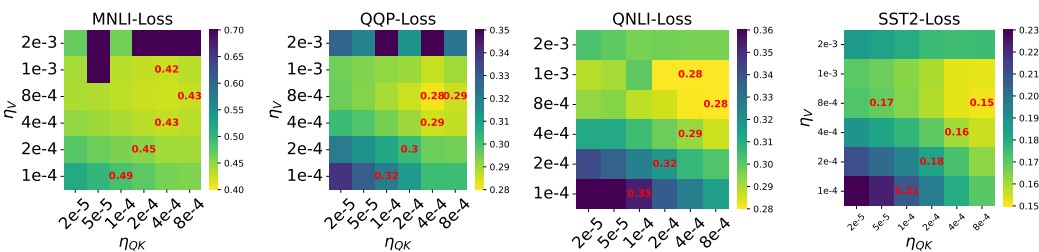

Figure 3: The train loss of RoBERTa-base fine-tuning. Other settings are same to Figure 1.

### C.2.2 MNLI Llama3.1-8b

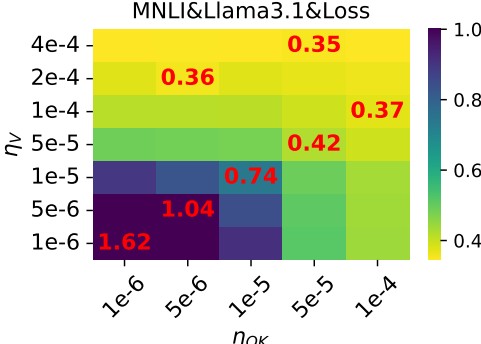

Figure 4: The train loss of Llama3.1-8b fine-tuning. Other settings are same to Figure 2.

### C.2.3 Algorithm Framework

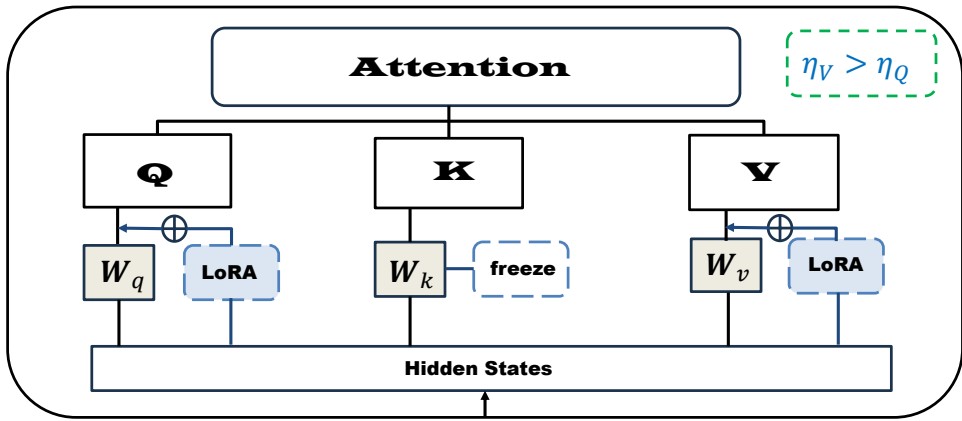

Figure 5: A brief diagram outlining how our theoretical insights guide the experiments.

### C.2.4 ABLATION EXPERIMENTS ON MISTRAL-7B

In alignment with the experimental setup (hyperparameter setting for Llama3.1-8b) described in our Section 4.1, Figure 2, we have evaluated the RTE and MNLI task performances of our approach on Mistral-7B:

| Method | Hyperparameter | RTE | MNLI |
|---|---|---|---|
| LoRA (QKV) | $r = 16, \lambda = 1$ | 81.28 | 87.80 |
| LoRA (QV) | $r = 16, \lambda = 1$ | 80.51 | 88.87 |
| LoRA (QV) | $r = 16, \lambda = 2$ | 81.59 | **89.04** |
| LoRA (QV) | $r = 16, \lambda = 4$ | 80.87 | 88.64 |
| LoRA (QV) | $r = 16, \lambda = 8$ | **83.75** | 88.78 |

### C.2.5 MORE CHALLENGING EVALUATION

Evaluating the model on more challenging benchmarks is essential for a comprehensive understanding of its capabilities. To address this, we follow [52] to fine-tune the LLaMA3.1-8B model on the MetaMathQA [52] dataset (the training set consists of the first 10K samples selected from the 150K MetaMathQA dataset.) and evaluate the performance on the GSM8K [6] (a benchmark for mathematical problem-solving).

| Method | GSM8K (100%) |
|---|---|
| Before fine-tune | 25.55 |
| LoRA (QKV) $r = 16, \lambda = 1$ | 57.70 |
| LoRA (QV) $r = 16, \lambda = 2$ | **59.15** |
| LoRA (QV) $r = 16, \lambda = 4$ | 58.23 |

### C.2.6 FINE-TUNING K,V

We fine-tune only $\mathbf{W}_k$ and $\mathbf{W}_v$ of Roberta-base, others are the same to Table 1.

| | Weight Type | $\mathbf{W}_k, \mathbf{W}_v$ | $\mathbf{W}_q, \mathbf{W}_v$ | $\mathbf{W}_q, \mathbf{W}_k, \mathbf{W}_v$ |
|---|---|---|---|---|
| **SST2**(R) | $r = 8$ | 0.920 | 0.919 | 0.922 |
| | $r = 16$ | 0.920 | 0.921 | 0.923 |
| **QNLI**(R) | $r = 8$ | 0.887 | 0.889 | 0.895 |
| | $r = 16$ | 0.888 | 0.890 | 0.890 |
| **QQP**(R) | $r = 8$ | 0.840 | 0.840 | 0.844 |
| | $r = 16$ | 0.840 | 0.839 | 0.844 |
| **MNLI**(R) | $r = 8$ | 0.821 | 0.820 | 0.827 |
| | $r = 16$ | 0.822 | 0.824 | 0.828 |

### C.2.7 Directly fine-tuning Q,K,V with lambda

We fine-tuning $\mathbf{W}_q, \mathbf{W}_k, \mathbf{W}_v$ with $\lambda$ directly with the same settings in Table 2 for easy comparison, supporting one of our major claims in Theorem 2.

| Method | Trainable #Param (M) | RTE | STS-B | MRPC | CoLA |
|---|---|---|---|---|---|
| Before Fine-tune | 0 | 45.12 | -3.18 | 66.66 | 1.09 |
| LoRA (QKV) $r = 8, \lambda = 1$ | 1.62 | 70.76 | 90.25 | 85.04 | 58.03 |
| LoRA (QKV) $r = 8, \lambda = 2$ | 1.62 | 72.92 | 90.54 | 86.76 | 58.28 |
| LoRA (QKV) $r = 8, \lambda = 4$ | 1.62 | 73.64 | 90.84 | 87.74 | **60.66** |
| LoRA (QKV) $r = 8, \lambda = 8$ | 1.62 | 76.10 | **91.00** | **88.48** | 60.59 |
| LoRA (QKV) $r = 16, \lambda = 1$ | 2.07 | 70.39 | 90.25 | 86.03 | 58.04 |
| LoRA (QKV) $r = 16, \lambda = 2$ | 2.07 | 72.56 | 90.36 | 86.27 | 59.81 |
| LoRA (QKV) $r = 16, \lambda = 4$ | 2.07 | 74.00 | 90.84 | 86.76 | 60.07 |
| LoRA (QKV) $r = 16, \lambda = 8$ | 2.07 | **76.97** | 90.81 | 87.74 | 60.34 |

