# OpenReview forum: "Theoretical Insights into Fine-Tuning Attention Mechanism: Generalization and Optimization"
_ICLR.cc/2025/Conference — Submitted to ICLR 2025_

### Official Review · Reviewer_TXeN · 2024-10-28

**Soundness:** 1
**Presentation:** 2
**Contribution:** 2
**Rating:** 3
**Confidence:** 3

**Summary:**

The paper investigates two phenomena observed during the fine-tuning of Transformer LLMs, focusing on the attention mechanism. The first phenomenon is the "Different Impact," where optimizing the Wv (value) matrix significantly improves performance more than optimizing the Wk (key) matrix. The second is "Efficient Convergence," where using distinct learning rates for different matrices is crucial for optimal performance, with a higher learning rate for Wv being beneficial for faster convergence. The paper provides theoretical analysis on these phenomena from the perspectives of generalization and optimization. Based on this, this paper proposes only finetuning Wq and Wv, and with a larger learning rate for Wv.

**Strengths:**

The observation that fine-tuning only Wq and Wv​ matrices yields performance gains comparable to finetuning of Wq, Wk and Wv is interesting. Trying to analyze and uncover the reason can benefit the research commnity.

**Weaknesses:**

See my questions.

**Questions:**

1. My biggest concern with this paper is that the key observation seems to originate from the LoRA paper: *"Adapting both \( W_q \) and \( W_v \) gives the best performance overall"* (this sentence is quoted from the LoRA paper), and even the experiment setup is similar to Table 5 in the LoRA paper. Deriving key observations from other works isn’t an issue (or saying we share the same observation/insights), but I believe it should be directly and explicitly pointed out, rather than presented as your own observation, concerning the lora paper is very famous for several years and you do cite it. Or could you clarify what differences there are between your work and the LoRA paper? Please provide a detailed comparison between their findings and those in the LoRA paper, highlighting any novel contributions or deeper insights their work provides. This would help clarify the paper's originality and contribution to the field.

2. As you pointed out on line 205, \( W_q \) and \( W_k \) can be treated as a single unit. In linear algebra, two matrices multiplied without an intermediate activation can be equivalent to a single matrix. This could explain why fine-tuning only \( W_q \) and \( W_v \) achieves comparable accuracy to tuning all matrices. If this is the case, have you tried fine-tuning only \( W_k \) and \( W_v \)? It seems that neither the LoRA paper nor your work has conducted this experiment. Could you please include this experiment in your study or explain why not to perform it. This could provide valuable insights and strengthen the paper's analysis.

3. What are the main conclusions of the theoretical analyses in Sections 3 and 4? I found it difficult to follow your proofs. Please include a clear summary of the main conclusions from their theoretical analyses at the end of Sections 3 and 4. I would suggest to provide more intuitive explanations to help readers better understand the theoretical analyses and proofs.

4. Section 5 appears to be simply fine-tuning \( W_q \) and \( W_v \) with a search for \lambda. Do you have any insights on how to determine \lambda? could you provide guidelines or heuristics for determining an appropriate \lambda value based on your theoretical and empirical results?

---

> ### Author Response · Authors · 2024-11-16
> **Thank you!**
>
> Dear Reviewer TXeN,
>
> We are grateful for your detailed review and the valuable insights you have shared. Your comments are highly appreciated, and we have addressed each point in the following responses.
>
> >**Q1:**  My biggest concern with this paper is that the key observation seems to originate from the LoRA paper: *"Adapting both ( $W_q$ ) and ( $W_v$ ) gives the best performance overall"* (this sentence is quoted from the LoRA paper), and even the experiment setup is similar to Table 5 in the LoRA paper. Deriving key observations from other works isn’t an issue (or saying we share the same observation/insights), but I believe it should be directly and explicitly pointed out, rather than presented as your own observation, concerning the lora paper is very famous for several years and you do cite it. Or could you clarify what differences there are between your work and the LoRA paper? Please provide a detailed comparison between their findings and those in the LoRA paper, highlighting any novel contributions or deeper insights their work provides. This would help clarify the paper's originality and contribution to the field.
>
> >**A:**  Thank you for giving us the opportunity to clarify our contributions. The key differences between our work and the LoRA paper are detailed as follows.
> >
> >(1) **Different experimental setup.**
> >
> >- In the LoRA paper (Section 7.1, Table 5), the authors "apply LoRA to different types of attention weights, **given the same number of trainable parameters** (Trainable Parameters = 18M)" — quoted from the LoRA paper.
> >
> >  Thus, their conclusion that "adapting both ($W_q$) and ($W_v$) gives the best performance overall" is drawn under the constraint of having the same total trainable parameters.
> >  (1. $W_q,W_v$ with rank 4; 2. $W_q,W_k,W_v,W_o$ with rank 2)
> >
> >- In contrast, our study, as presented in Table 1, compares fine-tuning different types of attention weights **under the same fine-tuning rank values** (each row in Table 1 illustrates this comparison).
> >
> >  Based on this setup, one of our key insights is that, when using the same rank rrr, fine-tuning only the $W_q$ and $W_v$ matrices is not only computationally more efficient but also delivers results comparable to, or even surpassing, fine-tuning all three matrices ($W_q$, $W_k$, and $W_v$).
> >
> >(2) **Different conclusions.**
> >
> >- Empirically, in LoRA paper, given the same number of trainable parameters, "it is preferable to adapt more weight matrices than adapting a single type of weights with a larger rank". (this sentence is quoted from the LoRA paper)
> >
> >- In our paper, given the same rank value in fine-tuning. (a) Empirically, fine-tuning both Wq and Wv often leads to **performance that matches or even exceeds** that achieved by fine-tuning all three matrices $W_q$, $W_k$, and $W_v$.  (b) Theoretically, in **Section 3.2 (Thereom 1)**, we provide generalization bounds for adapting $W_q, W_v$ and/or $W_k$. Specifically, fine-tuning only $W_q$ and $W_k$ reduces the number of parameters for the same rank rrr, while improving generalization bounds and offering potential memory advantages.
> >
> >(3) This, however, represents just a fraction of our contributions detailed in Section 3. (For further details, please refer to our responses **A** through **Q3**.)

---

> > ### Comment · Reviewer_TXeN · 2024-11-27
> > **Really Different Conclusions？**
> >
> > I believe your response is somewhat misleading. While you emphasize differences in the experimental settings, the key observation remains the same. It seems you are attempting to use this sentence: **"it is preferable to adapt more weight matrices than adapting a single type of weights with a larger rank,"** to obscure the fact that the more specific insight, **"Adapting both ($W_q$) and ($W_v$) gives the best performance overall,"** from the lora paper. This latter insight is simply a deeper extension of the former, and it appears to be same as your findings as well.
> >
> > So, you changed the setting (actually the setting difference is trivial, rank also means more parameter, just a fixed rank/parameter budget v.s. not-fixed budget) and then you arrived at the same conclusion? I believe this is not appropriate to present it as your own novel observation.

---

> > > ### Author Response · Authors · 2024-11-27
> > > **Further clarification on the contributions (theoretical)!**
> > >
> > > Dear Reviewer TXeN,
> > >
> > > Thank you for your detailed review and feedback on our work. We appreciate your points and the suggestions you have provided. We believe that, although there are adjustments in the experimental settings, **our conclusions are not merely a repetition of existing work (LoRA), but rather a significant theoretical extension**, which we would like to clarify here.
> > >
> > > >**(1)** Our experiment differs from the original LoRA setup, **as you acknowledge, in terms of "fixed rank/parameter budget vs. non-fixed budget."**  However, one of our main extended conclusions is that adapting only（$W_q$） and （$W_v$）can enhance generalization bounds and potentially offer memory benefits  with the same rank (r). (See **(2)** for details)
> > >
> > > >**(2) The main contribution of this work.**
> > > >
> > > >Our primary objective is to provide a general **theoretical framework** that reveals the underlying mechanism behind the  phenomena observed during the fine-tuning of LLMs involving the attention mechanism. Building on our experimental and theoretical insights, **one can develop new algorithms** to improve the effectiveness (e.g., storage, and time) of fine-tuning.
> > > >
> > > >According to the traditional statistical learning viewpoint, performance can be defined by the sum of **optimization error and generalization error**. Our theoretical analyses in Sections 3 and 4 correspond to generalization and optimization, respectively.
> > > >
> > > >- In Section 3 (generalization), we give **Thereom 1** (Information-theoretic genralization bounds), showing that **with the same r value**, fine-tuning $W_q,W_v$ consistently achieves results comparable to or even surpassing those of fine-tuning $W_q,W_k,W_v$. This reduces the number of parameters for the same r, while improving generalization bounds and potentially providing memory benefits.  (**storage-friendly**)
> > > >
> > > >
> > > >- In Section 4 (optimization), we discuss the learning dynamics in fine-tuning attention mechanism, and we illustrate (**Theorem 2**) that the feature learning of attention mechanism is efficient when the learning rate for $W_v$ should be generally much larger than that of $W_q,W_k$ in fine-tuning. (**time-friendly**)
> > > >
> > > >Building on our experimental and theoretical insights, example in Section 5 is presented solely to illustrate how theoretical analysis can guide experimental procedures effectively.
> > >
> > > >**(3)**
> > > >In summary, although our research shares similarities with the conclusions of LoRA (**We appreciate your feedback and will revise this presentation detail in the new version accordingly**), we believe that we have provided a more detailed theoretical analysis and experimental validation, along with a theoretical extension based on these foundations. We believe these contributions are substantial enough to be presented as an independent work, rather than a mere repetition of existing ideas.
> > >
> > > Once again, we thank you for your careful attention to our work. We will further clarify these details in the manuscript to ensure that our contributions are fully understood.
> > >
> > > Best regards.

---

> ### Author Response · Authors · 2024-11-16
>
> >**Q2:**  As you pointed out on line 205, ($W_q$ ) and ( $W_k$ ) can be treated as a single unit. In linear algebra, two matrices multiplied without an intermediate activation can be equivalent to a single matrix. This could explain why fine-tuning only ( $W_q$ ) and ( $W_v$ ) achieves comparable accuracy to tuning all matrices. If this is the case, have you tried fine-tuning only ( $W_k$ ) and ( $W_v $)? It seems that neither the LoRA paper nor your work has conducted this experiment. Could you please include this experiment in your study or explain why not to perform it. This could provide valuable insights and strengthen the paper's analysis.
>
> >**A:** That is a great question!
> >
> >(1) As you mentioned, in linear algebra, two matrices multiplied without an intermediate activation can be equivalent to a single matrix. Therefore, the effects of fine-tuning $W_q,W_v$ and $W_k,W_v$ should be similar. Please review the supplementary experimental results (#) provided below. (We fine-tune only $W_k$  and  $W_v $ of Roberta-base, others are the same to **Table 1**).
> >
> >| Task        | $ W_k,W_v$ # | $W_q,W_v$ | $W_q,W_k,W_v$ |
> >| ----------- | ------------ | --------- | ------------- |
> >| SST2 (r=8)  | 0.920        | 0.919     | **0.922**     |
> >| SST2 (r=16) | 0.920        | 0.921     | **0.923**     |
> >| QNLI (r=8)  | 0.887        | 0.889     | **0.895**     |
> >| QNLI (r=16) | 0.888        | **0.890** | 0.890         |
> >| QQP (r=8)   | 0.840        | 0.840     | **0.844**     |
> >| QQP (r=16)  | 0.840        | 0.839     | **0.844**     |
> >| MNLI (r=8)  | 0.821        | 0.820     | **0.827**     |
> >| MNLI (r=16) | 0.822        | 0.824     | **0.828**     |
> >
> >(2) Explain why not to perform it (neither the LoRA paper nor our work)?
> >
> >In Eq. (1) in Section 2, the conventional attention function includes a term $xW_qW_k^T$,  $W_k$ can only exploit the transformed representation matrix $xW_q$ generated by the preceding transformation, thereby losing direct access to the original representation information.
> >
> >This observation aligns with the insights in [1]: the information representation in LoRA also exhibits significant limitations, as in $\Delta h = xAB$, where $B$ similarly lacks access to the original representation information.
> >
> >[1] Haotong Qin et al. Accurate LoRA-Finetuning Quantization of LLMs via Information Retention. ICML 2024.

---

> ### Author Response · Authors · 2024-11-16
>
> >**Q3:** What are the main conclusions of the theoretical analyses in Sections 3 and 4? I found it difficult to follow your proofs. Please include a clear summary of the main conclusions from their theoretical analyses at the end of Sections 3 and 4. I would suggest to provide more intuitive explanations to help readers better understand the theoretical analyses and proofs.
>
> >**A:** According to the traditional statistical learning viewpoint, performance can be defined by the sum of **optimization error and generalization error**. Our theoretical analyses in Sections 3 and 4 correspond to generalization and optimization, respectively.
> >
> >In Section 3 (generalization), we give **Thereom 1** (Information-theoretic genralization bounds), showing that with the same r value, fine-tuning $W_q,W_v$ consistently achieves results comparable to or even surpassing those of fine-tuning $W_q,W_k,W_v$. This reduces the number of parameters for the same r, while improving generalization bounds and potentially providing memory benefits.  (**storage-friendly**)
> >
> >In Section 4 (optimization), we discuss the learning dynamics in fine-tuning attention mechanism, and we illustrate (**Theorem 2**) that the feature learning of attention mechanism is efficient when the learning rate for $W_v$ should be generally much larger than that of $W_q,W_k$ in fine-tuning. (**time-friendly**)
> >
> >Building on our experimental and theoretical insights, one **can develop new algorithms** to improve the effectiveness (e.g., storage, and time) of fine-tuning (Example in Section 5).
>
> >**Q4:** Section 5 appears to be simply fine-tuning ( $W_q$ ) and ( $W_v$ ) with a search for $\lambda$. Do you have any insights on how to determine $\lambda$? could you provide guidelines or heuristics for determining an appropriate $\lambda$ value based on your theoretical and empirical results?
>
> >**A:**  As discussed in **Remark 4**, the optimal ratio $\lambda$ depends on the architecture and the fine-tuning task via the constants in $\Theta$ in **Theorem 2**.  This is a limitation of these asymptotic results since they do not offer any insights on how the constants are affected by the task and the neural architecture.
> >
> >However, we can still employ some heuristic methods, such as: we can select an appropriate range by conducting a certain amount of experiments, as shown in **Figure 1**, it seems that a ratio of order $2^1-2^4$ is optimal. （That is why we do a simple search for $\lambda$ (2,4,8) in Section 5.）
> >
> >Moreover, $\lambda$ should not be too large; otherwise, as shown in the MNLI subplot in Figure 1, the model's performance will collapse ($\eta_{QK}=4e^{-4},\eta_V=2e^{-3}$).

---

> ### Author Response · Authors · 2024-11-26
> **We await your reply with anticipation!**
>
> Dear Reviewer TXeN,
>
> We sincerely appreciate your time and effort in reviewing our manuscript and providing valuable feedback.
>
> As the day that authors may upload a revised PDF phase nears completion, we wish to confirm whether our responses have effectively addressed your concerns. We provided detailed responses to your concerns a few days ago and hope they have adequately resolved any issues. If you require further clarification or have any additional concerns, please do not hesitate to contact us. We are more than willing to continue our communication with you.
>
> Best regards.

---

> ### Author Response · Authors · 2024-11-30
> **Once again, thank you so much!**
>
> Dear Reviewer TXeN,
>
> As we approach the rebuttal discussion deadline, we would like to follow up with you regarding the additional ablation experiments, theoretical (generalization and optimization) analysis, and writing updates we provided. Have these addressed your concerns? If so, we kindly ask if you would consider revising your review score accordingly.
>
> Thank you for your time and thoughtful evaluation of our work.
>
> Best regards.

---

### Official Review · Reviewer_f8uV · 2024-10-30

**Soundness:** 1
**Presentation:** 2
**Contribution:** 2
**Rating:** 5
**Confidence:** 4

**Summary:**

This paper explores fine-tuning strategies for the attention mechanism from a theoretical perspective, focusing on generalization and optimization issues, and claims to provide theoretical insights to guide algorithm design. It has two major propositions, one is fine-tuning WqWv matrix is more efficient and can achieve comparable results as fine-tuning WqWkWv matrix. Another is fine-tuning Wq and Wv should use different learning rates for more efficient convergence.

**Strengths:**

The idea of using different local learning rates for Wq and Wv is interesting. As V is timed with attention logits, q and v should have more different gradient distributions than q and k. Using different local learning rates for these two types of matrices is reasonable and worth exploring.

**Weaknesses:**

1. The writing and presentation are rather poor. Both sentence level and logical level polishment are suggested for this work.
    E.g. In line 16~22, "In this paper, we investigate two remarkable phenomena ... with a higher learning rate for the Wv matrix expediting convergence." The ordering of the sentences and the way of structuring the arguments add unnecessary difficulty to reading. This kind of problem happens across the whole manuscript.
    A suggestion is, use multiple statement sentences instead of clauses. A rewritten version of these sentences is " In this paper, we investigate two remarkable phenomena associated with attention mechanism, during the fine-tuning of LLMs. One is 'Different Impact' and another is 'Efficient Convergence'. ......" Another point is that the naming of "Different Impact" and "Efficient Convergence" doesn't help with understanding. Consider other ways to name them.

2. For the "Different Impact", the authors claimed that fine-tuning Wq&Wv is favored over Wq,Wk,Wv together. However, this argument is not supported by their experimental results in Table 1. It can be seen from the table that fine-tuning Wq,Wk,Wv together still gets the best performance for most of the cases. I wonder if the authors are writing the arguments and conducting experiments separately without any discussion. The authors should summarize their findings according to the experiment results.

**Questions:**

1.  In line 160~193, the author writes "As seen in the table, we can see a clear trend where solely updating the Wv matrix outperforms just learning the Wq,Wk matrix. Interestingly, the combination of fine-tuning both Wq and Wv often leads to performance that matches or even exceeds that achieved by fine-tuning all three matrices Wq,Wk, and Wv". This statement is not supported by the results in Table 1, as solely updating the Wv is inferior to updating Wq,Wk together, and fine-tuning all three matrices Wq,Wk, and Wv achieves the best result in most cases. The authors should first rewrite the whole section of "Different Impact" to have a consistent argument with the experiment results, and investigate more tasks to see whether there are certain types of tasks in which "different impact" is true.

---

> ### Author Response · Authors · 2024-11-16
> **Thanks!**
>
> Dear Reviewer f8uV,
>
> Thank you for taking the time to review our work and for offering thoughtful comments and constructive questions. Your feedback is instrumental in refining the manuscript. Please see our detailed replies below.
>
> >**W1：** The writing and presentation are rather poor. Both sentence level and logical level polishment are suggested for this work. E.g. In line 16~22, "In this paper, we investigate two remarkable phenomena ... with a higher learning rate for the Wv matrix expediting convergence." The ordering of the sentences and the way of structuring the arguments add unnecessary difficulty to reading. This kind of problem happens across the whole manuscript. A suggestion is, use multiple statement sentences instead of clauses. A rewritten version of these sentences is " In this paper, we investigate two remarkable phenomena associated with attention mechanism, during the fine-tuning of LLMs. One is 'Different Impact' and another is 'Efficient Convergence'. ......" Another point is that the naming of "Different Impact" and "Efficient Convergence" doesn't help with understanding. Consider other ways to name them.
>
> >**A：** We appreciate your feedback regarding the clarity and structure of the writing, particularly with respect to sentence flow and logical organization. In response, we will revise the relevant sections to simplify the sentence structures and ensure that the arguments are presented in a clearer and more direct manner. Below is the preliminary version of the revised text.
> >
> >In this paper, we investigate two notable phenomena related to the attention mechanism during the fine-tuning of large language models (LLMs). The first phenomenon, termed "Effect of Weight Combination on Performance," highlights the impact of fine-tuning different weight matrices. It demonstrates that optimizing the Wv matrix leads to a more significant performance improvement compared to optimizing the Wk matrix. Fine-tuning only the Wq and Wv matrices is not only computationally more efficient but also yields results comparable to, or even better than, fine-tuning all three matrices (Wq, Wk, and Wv). The second phenomenon, "Impact of Learning Rate on Convergence," emphasizes the crucial role of distinct learning rates for these matrices in achieving optimal performance, with a notably higher learning rate for the Wv matrix compared to Wq and Wk, which accelerates convergence.

---

> ### Author Response · Authors · 2024-11-16
>
> >**W2：** For the "Different Impact", the authors claimed that fine-tuning Wq&Wv is favored over Wq,Wk,Wv together. However, this argument is not supported by their experimental results in Table 1.  It can be seen from the table that fine-tuning Wq,Wk,Wv together still gets the best performance for most of the cases. **I wonder if the authors are writing the arguments and conducting experiments separately without any discussion. The authors should summarize their findings according to the experiment results.**
> >
> >**Q1：** In line 160~193, the author writes "As seen in the table, we can see a clear trend where solely updating the Wv matrix outperforms just learning the Wq,Wk matrix. Interestingly, the combination of fine-tuning both Wq and Wv often leads to performance that matches or even exceeds that achieved by fine-tuning all three matrices Wq,Wk, and Wv". **This statement is not supported by the results in Table 1, as solely updating the Wv is inferior to updating Wq,Wk together, and fine-tuning all three matrices Wq,Wk, and Wv achieves the best result in most cases.** The authors should first rewrite the whole section of "Different Impact" to have a consistent argument with the experiment results, and investigate more tasks to see whether there are certain types of tasks in which "different impact" is true.
>
> >**A:**  Regarding your comment on the "Different Impact" phenomenon and the results presented in Table 1, we believe there may have been a **misunderstanding in interpreting the data**. Please allow us to clarify the experimental results and our arguments.
> >
> >**Specifically, in our original statement:**
> >
> >(1) *“We can see a clear trend where solely updating the Wv matrix outperforms just learning the Wq,Wk matrix”* .
> >
> >(See the third and fourth columns)
> >
> >(2) *“Interestingly, the combination of fine-tuning both Wq and Wv often leads to **performance that matches or even exceeds** that achieved by fine-tuning all three matrices Wq, Wk, and Wv”*.
> >
> >(See the fifth and sixth columns)
> >
> >**In your feedback:**
> >
> >(1) "as solely updating the Wv is inferior to updating Wq,Wk together. "
> >
> >Compare the third (Wv) and fourth (Wq,Wk together) columns of the Table 1, the values in the third column are generally superior to those in the fourth column.
> >
> >(2) "fine-tuning all three matrices Wq,Wk, and Wv achieves the best result in most cases."
> >
> >This statement is correct, but our **main point** is that the performance is comparable to, and in some cases may even exceed (especially fine-tuning Llama3.1-8b on QQP, MNLI), rather than being the best.
>
> Once again, thank you for your valuable feedback. We hope our clarification addresses your concerns and resolves any possible misunderstandings.

---

> > ### Comment · Reviewer_f8uV · 2024-11-17
> >
> > As the authors respond actively, I would like to provide further feedback.
> >
> > For the previously mentioned w1, I suggest using terms like "unequal importance of attention matrices" and "attention matrices with customized learning rate leads to better convergence". These two phenomena both come from the attention equation itself, and it is worth mentioning it as a motivation. The current discussion of this kind of motivation is in sec 4, and I suggest the authors adjust the logical flow. The authors should put a reasonable amount of motivational discussion in the introduction instead of just repeating the abstract.
> >
> > For the rebuttal phase, a common way to show the modification/improvements is to modify the paper content with colored text. Only when I see enough improvement in the paper quality, I will improve my score on the writing part.
> >
> > And for w2, thanks for pointing out it is comparing fine-tuning wv and fine-tuning (wk, wq) in the first half of the sentence, I misread this half of the sentence. The authors should be careful with the page layout here, this sentence crosses the page with a table in the middle.
> >
> > As the authors agree, fine-tuning wq, wk, wv still gives the best performance in most cases. In the later experiments in Table 2, I would expect fine-tuning wq, wk, wv with $\lambda$ will give the best results for both full fine-tuning and LoRA setting, while it is not shown in the table. For the claim of efficient convergence, it is better also to show wall clock time improvements and peak memory cost (if possible). For the current two major claims of this work, the authors need more ablations to demonstrate that fine-tuning only wq wv is an acceptable choice (either we can do this on cheaper GPU, or the efficiency improvement over performance drop is a reasonable trade-off). Another ablation is on rank choice over matrix number for LoRA fine-tuning: will increasing rank for wq,wv be favored over fine-tuning wq,wk,wv? This may potentially show the benefit of fine-tuning only wq and wv.
> >
> > And I would suggest it is worth exploring how to choose good $\lambda$ values from both theoretical and empirical aspects. The current grid of $\lambda$ still needs search. This could potentially lead to a higher research impact.

---

> > > ### Author Response · Authors · 2024-11-18
> > > **Glad to hear back from you!**
> > >
> > > Dear Reviewer f8uV,
> > >
> > > We sincerely appreciate and are truly honored to receive your further feedback. Please review our further responses to your valuable suggestions. To provide a detailed response, we have labeled your further feedback as **F1** to **F7**. We are addressing your points step by step.
> > >
> > > > **F1:** For the previously mentioned w1, I suggest using terms like "unequal importance of attention matrices" and "attention matrices with customized learning rate leads to better convergence". These two phenomena both come from the attention equation itself, and it is worth mentioning it as a motivation. The current discussion of this kind of motivation is in sec 4, and I suggest the authors adjust the logical flow. The authors should put a reasonable amount of motivational discussion in the introduction instead of just repeating the abstract.
> > > >
> > > > (Revision line 43 - line 65)
> > > >
> > > > **A:** We are pleased to adopt your suggestions. Specifically, we have **revised the terms as you proposed and adjusted the logical flow of the introduction** accordingly. Please kindly review the **updated revision**.
> > > >
> > > > We would like to add that we had already discussed **Case 1 from Section 4** in the original introduction as a motivation, and we have further refined the logical flow for better clarity.
> > >
> > > > **F2:** For the rebuttal phase, a common way to show the modification/improvements is to modify the paper content with colored text. Only when I see enough improvement in the paper quality, I will improve my score on the writing part.
> > > >
> > > > **A:** We have revised the paper according to your high-quality suggestions and highlighted the modifications with colored text, please kindly review our rebuttal revision.
> > >
> > > > **F3:** And for w2, thanks for pointing out it is comparing fine-tuning $w_v$ and fine-tuning ($w_k$, $w_q$) in the first half of the sentence, I misread this half of the sentence. The authors should be careful with the page layout here, this sentence crosses the page with a table in the middle.
> > > >
> > > > **A:** We are glad to hear that our clarification addressed your misreading. Thank you for bringing up the feedback, and we take extra care with the page layout to ensure a smoother read experience. (The sentence is now placed between line 167 and line174 without any table in between!)
> > >
> > > > **F4:** As the authors agree, fine-tuning $w_q, w_k, w_v$ still gives the best performance in most cases. In the later experiments in Table 2, I would expect fine-tuning $w_q, w_k, w_v$ with $\lambda$ will give the best results for both full fine-tuning and LoRA setting, while it is not shown in the table. For the claim of efficient convergence, it is better also to show wall clock time improvements and peak memory cost (if possible).
> > > >
> > > > **A:** We greatly appreciate your high-level scientific perspective. Please allow us to point out the **existing experiments that address the aspects you have highlighted** in your comments.
> > > >
> > > > (1) Fine-tuning $w_q,w_k,w_v$ with $\lambda$.
> > > >
> > > > When we present the second interesting phenomenon, *Efficient Convergence*, in **Section 4.1 (Figures 1 and 2)**, we are indeed referring to fine-tuning $w_q, w_k, w_v$ with $\lambda$ in LoRA setting. Moreover, as stated in the main text, additional empirical details are provided in Appendix C.1.
> > > >
> > > > (2) Time improvements.
> > > >
> > > > Also in **Section 4.1 (Figure 2 Right)**, we present the training loss over 800 steps for MNLI fine-tuning on Llama3.1-8b, showing comparison between two learning rate settings in Figure 2 Left : （a) $\eta_{QK}=\eta_{V}=1e^{-4}$  (2) $\eta_{QK}=5e^{-6},\eta_{V}=2e^{-4}$
> > > >
> > > > Under the same number of training steps, setting (b) achieves faster loss reduction compared to setting (a). This serves as the foundation for the claim of *Efficient Convergence*.
> > > >
> > > > (3) Memory cost.
> > > >
> > > > In **Section 3.1**, we present the **computational benefits** of tuning only $W_q$ and $W_v$, and in **Section 5 (Table 2)**, we demonstrate the training parameters for different fine-tuning methods, comparing $W_q, W_v$ and $W_q, W_k, W_v$.

---

> > > ### Author Response · Authors · 2024-11-18
> > >
> > > > **F5:** For the current two major claims of this work, the authors need more ablations to demonstrate that fine-tuning only $w_q,w_v$ is an acceptable choice (either we can do this on cheaper GPU, or the efficiency improvement over performance drop is a reasonable trade-off).
> > > >
> > > > **A:**
> > > >
> > > > (1) For more ablations to demonstrate that fine-tuning only $w_q,w_v$ is an acceptable choice
> > > >
> > > > We have added ablation experiments on different fine-tuning methods (Dora [1]) and models (Mistral-7B [2]). Please refer to our supplementary experiments for details.
> > > >
> > > > Following the experimental setup (hyperparameter setting for Llama3.1-8b) described in our Section 5, Table 2, we conducted additional evaluations on DoRA: (Revision line 493- line 510)
> > > >
> > > > >| **Method**                  | **Trainable #Param (M)** | **RTE**   | **STS-B** | **MRPC** | **CoLA**  | **MNLI**  | **SST-2** | **QQP**   | **QNLI**  |
> > > > >| --------------------------- | ------------------------ | --------- | --------- | -------- | --------- | --------- | --------- | --------- | --------- |
> > > > >| DoRA (QKV) $r=8$            | 1.06                     | 70.75     | 90.39     | 85.78    | 56.79     | 86.73     | 93.58     | 89.34     | 92.22     |
> > > > >| DoRA (QKV) $r=16$           | 1.51                     | 70.40     | 90.31     | 86.03    | 57.81     | 86.77     | 93.92     | 89.30     | 92.48     |
> > > > >| DoRA (QV) $r=8, \lambda=2$  | 0.90                     | 71.12     | 90.29     | 87.01    | 58.54     | 87.08     | 93.96     | 89.60     | 92.60     |
> > > > >| DoRA (QV) $r=8, \lambda=4$  | 0.90                     | 75.45     | 90.82     | 86.76    | 60.32     | 86.98     | 93.81     | 90.33     | _92.97_   |
> > > > >| DoRA (QV) $r=8, \lambda=8$  | 0.90                     | 70.76     | 90.38     | 87.75    | 57.01     | **87.12** | 94.15     | 90.45     | 92.48     |
> > > > >| DoRA (QV) $r=16, \lambda=2$ | 1.20                     | 69.68     | 90.53     | 87.75    | 59.31     | 87.09     | 93.92     | 89.68     | 92.70     |
> > > > >| DoRA (QV) $r=16, \lambda=4$ | 1.20                     | 76.16     | 90.77     | 88.48    | **60.84** | 86.96     | 94.15     | 90.34     | **93.01** |
> > > > >| DoRA (QV) $r=16, \lambda=8$ | 1.20                     | **77.26** | 90.83     | 88.96    | 60.32     | 87.10     | **94.17** | **90.46** | 92.80     |
> > > >
> > > > In alignment with the experimental setup (hyperparameter setting for Llama3.1-8b) described in our Section 4.1, Figure 2, we have evaluated the RTE and MNLI task performances of our approach on Mistral-7B : (Revision line 927- line 983)
> > > >
> > > > | **Method** | Hyperparameter        | RTE       | MNLI      |
> > > > | ---------- | --------------------- | --------- | --------- |
> > > > | LoRA (QKV) | $r=16$，$\lambda=1$   | 81.28     | 87.80     |
> > > > | LoRA (QV)  | $r=16$,   $\lambda=1$ | 80.51     | 88.87     |
> > > > | LoRA (QV)  | $r=16$,   $\lambda=2$ | 81.59     | **89.04** |
> > > > | LoRA (QV)  | $r=16$,   $\lambda=4$ | 80.87     | 88.64     |
> > > > | LoRA (QV)  | $r=16$,   $\lambda=8$ | **83.75** | 88.78     |
> > > >
> > > > [1] Liu, Shih-Yang, et al.  Dora: Weight-decomposed low-rank adaptation.  ICML Oral 2024.
> > > >
> > > > [2] https://github.com/mistralai
> > > >
> > > > (2) For GPU, or the efficiency improvement
> > > >
> > > > We have confirmed the validity through **both theoretical analysis and experimental results**.
> > > >
> > > > Theoretically:
> > > >
> > > > According to the traditional statistical learning viewpoint, performance can be defined by the sum of **optimization error and generalization error**. Our theoretical analyses in Sections 3 and 4 correspond to generalization and optimization, respectively.
> > > >
> > > > In Section 3 (generalization), we give **Thereom 1** (Information-theoretic genralization bounds), showing that with the same r value, fine-tuning $W_q,W_v$ consistently achieves results comparable to or even surpassing those of fine-tuning $W_q,W_k,W_v$. This reduces the number of parameters for the same r, while improving generalization bounds and potentially providing memory benefits.  (**storage-friendly**)
> > > >
> > > > In Section 4 (optimization), we discuss the learning dynamics in fine-tuning attention mechanism, and we illustrate (**Theorem 2**) that the feature learning of attention mechanism is efficient when the learning rate for $W_v$ should be generally much larger than that of $W_q,W_k$ in fine-tuning. (**time-friendly**)
> > > >
> > > > Building on our experimental and theoretical insights, one **can develop new algorithms** to improve the effectiveness (e.g., storage, and time) of fine-tuning (Example in Section 5).
> > > >
> > > > Experimentally:
> > > >
> > > > Please revisit **A to F4** and **A (1) to F5** for the experimental validity responses.

---

> > > ### Author Response · Authors · 2024-11-18
> > >
> > > > **F6:** Another ablation is on rank choice over matrix number for LoRA fine-tuning: will increasing rank for $w_q,w_v$ be favored over fine-tuning $w_q,w_k,w_v$? This may potentially show the benefit of fine-tuning only $w_q$ and $w_v$.
> > > >
> > > > **A:** That is a great question! This is precisely what we aim to demonstrate in our whole paper (Theoretical Insights into Fine-Tuning Attention Mechanism: Generalization and Optimization). **Our strategy is primarily intended to demonstrate how theoretical analysis can effectively guide experimental procedures.**
> > > >
> > > > As clarified in **A to F5**, the core of our work lies in the theoretical analysis of the attention mechanism, providing effective theoretical guidance for the design of various methods. One of our key insights is that, **under the constraint of the same total trainable parameter budget**, we can allocate more parameters to $W_q$ and $W_v$. （as you mentioned）
> > > >
> > > > This is also why we conducted the rank choice ablation experiments presented in **Table 1 and Table 2**, where we compare different LoRA ranks.
> > >
> > > > **F7:** And I would suggest it is worth exploring how to choose good $\lambda$ values from both theoretical and empirical aspects. The current grid of $\lambda$ still needs search. This could potentially lead to a higher research impact.
> > > >
> > > > (Revision line 463-line 470)
> > > >
> > > > **A:** Good suggetion!
> > > >
> > > > Theoretically:
> > > >
> > > > As discussed in **Remark 4**, the optimal ratio $\lambda$ depends on the architecture and the fine-tuning task via the constants in $\Theta$ in **Theorem 2**.  This is a limitation of these asymptotic results since they do not offer any insights on how the constants are affected by the task and the neural architecture.
> > > >
> > > > Empirically:
> > > >
> > > > However, we can still employ some heuristic methods, such as: we can select an appropriate range by conducting a certain amount of experiments, as shown in **Figure 1**, it seems that a ratio of order $2^1-2^4$ is optimal. （That is why we do a simple search for $\lambda$ (2,4,8) in Section 5.）
> > > >
> > > > Moreover, $\lambda$ should not be too large; otherwise, as shown in the MNLI subplot in Figure 1, the model's performance will collapse ($\eta_{QK}=4e^{-4},\eta_V=2e^{-3}$).

---

> > > > ### Comment · Reviewer_f8uV · 2024-11-25
> > > >
> > > > Thanks to the authors for the updates.
> > > >
> > > > What I was suggesting is to put the results with fine-tuning wq, wk, wv with $\lambda$ directly in Table 2 for easy comparison, and also include these $\lambda$ tuning results in other comparisons. If the $\lambda$ factor is not dominantly beneficial in those settings, your major claim would be clearer. If it is more important then the major focus should be adjusted accordingly.
> > > >
> > > > In the rebuttal phase, the paper writing quality is improved to an extent, while I suggest the authors to further polish the work for a better presentation. I would like to increase my score to 5.

---

> > > > > ### Author Response · Authors · 2024-11-26
> > > > > **Thank you once again!**
> > > > >
> > > > > Dear Reviewer f8uV,
> > > > >
> > > > > Thank you once again for your thoughtful and constructive feedback. We truly appreciate your patience and the valuable guidance you have provided.
> > > > >
> > > > > We have incorporated the **fine-tuning results you mentioned into Section C.2.7** (fine-tuning $w_q, w_k, w_v$ with $\lambda$, supporting one of our major claims in Theorem 2) for better clarity and comparison, as per your suggestion. Additionally, we will make every effort to further polish the paper to enhance its presentation quality.
> > > > >
> > > > > If possible, we would sincerely appreciate it if you could reconsider adjusting the score to a positive one (5 is negative), as it would greatly encourage and support our continued efforts.
> > > > >
> > > > > Thank you for your support and encouragement.
> > > > >
> > > > > Best regards.

---

### Official Review · Reviewer_FTN2 · 2024-11-04

**Soundness:** 3
**Presentation:** 2
**Contribution:** 3
**Rating:** 6
**Confidence:** 3

**Summary:**

This paper presents theoretical and empirical insights into improving the efficiency of fine-tuning large language models. The key contributions are as follows:

1. Different Impact in Fine-Tuning: The paper demonstrates that fine-tuning only the value (`Wv`) and query (`Wq`) matrices of the attention mechanism is computationally efficient and can yield results comparable to or better than fine-tuning all three matrices (`Wq`, `Wk`, `Wv`). It suggests that `Wv` has a significantly greater impact on performance compared to `Wk`.

2. Efficient Convergence: The authors show that using distinct learning rates for different matrices, particularly a higher rate for `Wv`, expedites convergence and optimizes the fine-tuning process. This insight provides a practical approach to achieving better fine-tuning results with fewer computational resources.

3. Theoretical Analysis: The paper provides a theoretical analysis from the perspectives of generalization and optimization. It establishes that fine-tuning `Wq` and `Wv` enhances memory efficiency and generalization bounds while optimizing convergence dynamics by accelerating learning for specific matrices.

4. Proposed Strategy: Building on these insights, the authors propose a strategy to improve fine-tuning efficiency, both in terms of storage and time. Experimental results on benchmark datasets validate the effectiveness of this approach.

**Strengths:**

- Originality: The originality of this paper lies in its focused approach to fine-tuning the attention mechanism of large language models. By exploring the selective fine-tuning of `Wv` and `Wq` matrices, the authors introduce a novel method that challenges the conventional approach of fine-tuning all attention matrices (`Wq`, `Wk`, `Wv`). The theoretical insights into the distinct roles of these matrices, combined with empirical validation, provide a fresh perspective on optimizing the attention mechanism. Additionally, the proposed use of differentiated learning rates for convergence further demonstrates creativity in enhancing the fine-tuning process.

- Quality: The quality of the paper is reflected in its theoretical analysis and empirical validation. The authors employ both information-theoretic methods and optimization theory to support their claims, which significantly strengthens the credibility of their contributions.

- Clarity: The paper is generally well-written, making the complex ideas accessible to a broad audience.

- Significance: The significance of the paper lies in its potential to influence future research and practical implementations of fine-tuning large language models. This contribution is highly relevant for both academia and industry, as it provides a more resource-efficient approach to adapting pre-trained models for downstream tasks.

**Weaknesses:**

1. **Lack of Base Model Performance for Each Task**: The paper does not provide the performance of the base model before fine-tuning for each task, making it challenging to evaluate the true effectiveness of the fine-tuning methods. Including these baseline results would help contextualize the improvements made through fine-tuning.

2. **GLUE Evaluation Is Too Simple for LLaMA3.1-8B**: The use of the GLUE benchmark to evaluate the LLaMA3.1-8B model is insufficient, as GLUE tasks are relatively simple compared to the capabilities of such a large model. Evaluating on more challenging tasks, such as coding or mathematical problem, could make the results more convincing and demonstrate the robustness of the proposed approach.

3. **Lack of Ablation Studies**: The paper could benefit from more extensive ablation studies to isolate the effects of different components of the proposed method, such as the impact of learning rate scaling and the specific contribution of each matrix (`Wq`, `Wv`). This would provide a clearer understanding of the factors contributing to the observed performance gains.

4. **Lack of Guidance on Hyperparameter `λ`**: The paper does not provide sufficient guidance on how to choose the hyperparameter `λ`, which controls the relative learning rates of different matrices. Without explicit guidelines or a heuristic for selecting `λ`, practitioners may find it difficult to replicate the reported results or apply the method effectively in different contexts.

**Questions:**

1. **Baseline Performance**: Could you provide the performance metrics of the base model before fine-tuning for each of the tasks? This would help in better understanding the relative improvements brought by your fine-tuning strategy.

2. **More Challenging Evaluations**: Have you considered evaluating the LLaMA3.1-8B model on more complex benchmarks, such as tasks involving coding or mathematical problem? Including such challenging tasks would make your results more comprehensive and convincing.

3. **Ablation Studies**: It would be helpful if you could add more ablation studies to isolate the effects of different components of the proposed method, such as the specific impact of `Wq` vs. `Wv` fine-tuning or the effect of learning rate scaling. This would provide a clearer picture of what drives the observed improvements.

4. **Hyperparamete `λ`**: Could you provide more guidance on how to choose the hyperparameter `λ`?

---

> ### Author Response · Authors · 2024-11-16
> **Thanks!**
>
> Dear Reviewer FTN2,
>
> Thank you for your careful evaluation and thoughtful suggestions. We deeply value your time and effort in providing such constructive feedback. Below, we address your comments point by point.
>
> >**Q1 (W1):** **Baseline Performance**: Could you provide the performance metrics of the base model before fine-tuning for each of the tasks? This would help in better understanding the relative improvements brought by your fine-tuning strategy.
>
> >**A:** The performance (100%) of the base model for each task is presented in the table below.
> >
> >Roberta-base：
> >
>
>
> >| RTE   | STS-B | MRPC  | CoLA | MNLI  | SST-2 | QQP   | QNLI  |
> >| ----- | ----- | ----- | ---- | ----- | ----- | ----- | ----- |
> >| 45.12 | -3.18 | 66.66 | 1.09 | 32.95 | 49.31 | 44.72 | 50.81 |
> >
> >Llama3.1-8b：(Notice that the MNLI task for Llama3.1-8b is still a complex task.)
> >
> >| QQP   | MNLI  |
> >| ----- | ----- |
> >| 55.08 | 33.34 |
>
>
>
> >**Q2 (W2):** **More Challenging Evaluations**: Have you considered evaluating the LLaMA3.1-8B model on more complex benchmarks, such as tasks involving coding or mathematical problem? Including such challenging tasks would make your results more comprehensive and convincing.
>
> >**A：** We completely agree that evaluating the model on more challenging benchmarks is essential for a comprehensive understanding of its capabilities. To address this, we follow [1] to fine-tune the LLaMA3.1-8B  model on the MetaMathQA [1]  dataset (the training set consists of the first 10K samples selected from the 150K MetaMathQA dataset.) and evaluate the performance on the GSM8K [2] (a benchmark for mathematical problem-solving).
> >
> >| Method                         | GSM8K (100%) |
> >| ------------------------------ | ------------ |
> >| Before fine-tune               | 25.55        |
> >| LoRA (QKV)   $r=16,\lambda=1$  | 57.70        |
> >| LoRA (QV)     $r=16,\lambda=2$ | 59.15        |
> >| LoRA (QV)     $r=16,\lambda=4$ | 58.23        |
> >
> >[1] Longhui Yu, et al. MetaMath: Bootstrap Your Own Mathematical Questions for Large Language Models. ICLR 2024.
> >
> >[2] Karl Cobbe, et al. "Training Verifiers to Solve Math Word Problems." arXiv preprint arXiv:2110.14168 (2021).
>
> >**Q3 (W3):** **Ablation Studies**: It would be helpful if you could add more ablation studies to isolate the effects of different components of the proposed method, such as the specific impact of `Wq` vs. `Wv` fine-tuning or the effect of learning rate scaling. This would provide a clearer picture of what drives the observed improvements.
>
> >**A:** Good suggestions!  Thank you for giving us the opportunity to **clarify the ablation studies presented in the main text**.
> >
> >(1) For the effects of different components ( such as `Wq` vs. `Wv` ).
> >
> >In **Section 3.1 (Table 1)**, we provide a detailed comparison of the impact of fine-tuning different weight matrices.
> >
> >We can see a clear trend where solely updating the $W_v$ matrix outperforms just learning the $W_q,W_k$ matrix. Interestingly, the combination of fine-tuning both $W_q$ and $W_v$ often leads to performance that matches or even exceeds that achieved by fine-tuning all three matrices $W_q,W_k,$ and $W_v$. This pattern is consistently observed across various tasks and rank values, further emphasizing the importance of these two matrices over $W_k$ during fine-tuning.
> >
> >(2) For the effect of learning rate scaling.
> >
> >In **Section 4.1 (Figure 1 and Figure 2)**, we compared the results of various learning rate combinations ($\eta_V,\eta_{QK}$).
> >
> >We observe that (1) test accuracy consistently reaches its maximum for certain sets of learning rates where $\eta_{QK}< \eta_V$,  outperforming the standard practice of setting $\eta_{QK}$ and $\eta_V$ equal. (2) More interestingly, the gap between the optimal choice of learning rates overall and the optimal choice when $\eta_{QK}=\eta_V$ varies across different tasks.
>
> >**Q4 (W4):** Hyperparamete `λ`: Could you provide more guidance on how to choose the hyperparameter `λ`?
>
> >**A:**  As discussed in **Remark 4**, the optimal ratio $\lambda$ depends on the architecture and the fine-tuning task via the constants in $\Theta$ in **Theorem 2**.  This is a limitation of these asymptotic results since they do not offer any insights on how the constants are affected by the task and the neural architecture.
> >
> >However, we can still employ some heuristic methods, such as: we can select an appropriate range by conducting a certain amount of experiments, as shown in **Figure 1**, it seems that a ratio of order $2^1-2^4$ is optimal. （That is why we do a simple search for $\lambda$ (2,4,8) in Section 5.）
> >
> >Moreover, $\lambda$ should not be too large; otherwise, as shown in the MNLI subplot in Figure 1, the model's performance will collapse ($\eta_{QK}=4e^{-4},\eta_V=2e^{-3}$).

---

> > ### Comment · Reviewer_FTN2 · 2024-11-27
> >
> > Thank you for your detailed response to my comments. I appreciate the time and effort you put into addressing the points I raised.
> >
> > After carefully considering your rebuttal, I have decided to increase my rating from 5 to 6.

---

> > > ### Author Response · Authors · 2024-11-27
> > > **Thank you again for your support and for your thorough review!**
> > >
> > > Dear Reviewer FTN2,
> > >
> > > Thank you very much for your thoughtful and constructive feedback. We greatly appreciate the time and effort you dedicated to reviewing our work.  We are also grateful for your positive evaluation and the increase in your rating. Your comments have been invaluable in refining our manuscript.
> > >
> > > Best regards.

---

> ### Author Response · Authors · 2024-11-26
> **We eagerly await your response!**
>
> Dear Reviewer FTN2,
>
> We sincerely appreciate your time and effort in reviewing our manuscript and providing valuable feedback.
>
> As the day that authors may upload a revised PDF phase nears completion, we wish to confirm whether our responses have effectively addressed your concerns. We provided detailed responses to your concerns a few days ago and hope they have adequately resolved any issues. If you require further clarification or have any additional concerns, please do not hesitate to contact us. We are more than willing to continue our communication with you.
>
> Best regards.

---

### Official Review · Reviewer_SWzh · 2024-11-04

**Soundness:** 3
**Presentation:** 3
**Contribution:** 3
**Rating:** 5
**Confidence:** 3

**Summary:**

This paper investigates and addresses the resource-intensive nature of fine-tuning Large Language Models (LLMs) on transformer architectures, focusing specifically on the attention mechanism. The study identifies two key phenomena:

1. **Different Impact**: The research demonstrates that optimizing the value matrix (Wv) in the attention mechanism leads to better performance improvements than optimizing the key matrix (Wk). Moreover, fine-tuning just the query (Wq) and value (Wv) matrices—not only reduces computational demands but also yields results that are on par with or surpass full matrix optimization (including Wq, Wk, Wv).

2. **Efficient Convergence**: The paper highlights the importance of employing distinct learning rates for different matrices, particularly using a higher learning rate for Wv to speed up the convergence process.

The contributions of the paper are both theoretical and practical. Theoretically, it offers a new analysis on how selective fine-tuning of the attention matrices can enhance generalization bounds and improve memory efficiency. Practically, it proposes a fine-tuning strategy that optimizes model training time and resource use. The effectiveness of this approach is supported by experimental results on benchmark datasets, which validate the proposed methods and lay the groundwork for developing more efficient algorithms for fine-tuning LLMs.

**Strengths:**

- **Importance of Understanding Attention Mechanisms During Fine-Tuning**: The challenge of gaining a deeper understanding of the attention mechanism during fine-tuning is a critical one. The approach developed in this paper has the potential to serve as a plug-and-play solution for achieving improved accuracy-efficiency trade-offs in LLM fine-tuning.

- **Empirical and Theoretical Contributions**: This paper offers both empirical and theoretical analyses to elucidate the behavior of the attention mechanism during fine-tuning. The insights provided here could make a valuable contribution to the field, supporting the development of enhanced fine-tuning techniques that optimize the accuracy-efficiency trade-off.

**Weaknesses:**

- **Generalizability of the Proposed Approach**: The primary concern is the generalizability of the proposed approach. Specifically, the authors could enhance the analysis by demonstrating that the proposed method consistently improves LLM performance across diverse scenarios. To this end, it would be beneficial to include performance results under more complex, open-ended generation tasks, such as MT-Bench or comparable challenging benchmarks. Additionally, considering variations in model behavior, evaluating the approach on a broader range of LLMs, such as Mistral, would further strengthen the claim of general applicability.

- **Visualization for Better Insight**: To deepen the analysis and understanding of the findings, visualizing the learned attention distributions across different settings could be valuable. By examining how attention distributions vary under different configurations, the authors could offer a more nuanced understanding of the observed effects, shedding light on the underlying phenomenon.

**Questions:**

- How will the proposed method perform on other PEFT techniques, such as DoRA [1]?

[1] Liu, Shih-Yang, et al. "Dora: Weight-decomposed low-rank adaptation." arXiv preprint arXiv:2402.09353 (2024).

---

> ### Author Response · Authors · 2024-11-16
> **Thank you!**
>
> Dear Reviewer SWzh,
>
> We sincerely appreciate your thorough review and the insightful comments you provided. Your feedback is invaluable in improving our paper. Please find our detailed response below.
>
>
>
> >**Q1:** How will the proposed method perform on other PEFT techniques, such as DoRA [1]?
> >
> >[1] Liu, Shih-Yang, et al. "Dora: Weight-decomposed low-rank adaptation." arXiv preprint arXiv:2402.09353 (2024).
>
> >**A:** Following the experimental setup described in our Section 5, Table 2, we conducted additional evaluations on DoRA. Below are the results of our supplementary experiments:
> >
> >| **Method**                      | **Trainable #Param (M)** | **RTE**   | **STS-B** | **MRPC**  | **CoLA**  | **MNLI**  | **SST-2** | **QQP**   | **QNLI**  |
> >| ------------------------------- | ------------------------ | --------- | --------- | --------- | --------- | --------- | --------- | --------- | --------- |
> >| Full Fine-tune (QKV)            | 21.85                    | 73.64     | 90.49     | 84.55     | 60.34     | 86.68     | 93.23     | _90.48_   | 92.37     |
> >| LoRA (QKV) $r=8$                | 1.62                     | 70.76     | 90.25     | 85.04     | 58.03     | 86.70     | 93.92     | 89.15     | 92.17     |
> >| LoRA (QKV) $r=16$               | 2.07                     | 70.39     | 90.25     | 86.03     | 58.04     | 86.78     | 93.92     | 89.26     | 92.18     |
> >| DoRA (QKV) $r=8$                | 1.06                     | 70.75     | 90.39     | 85.78     | 56.79     | 86.73     | 93.58     | 89.34     | 92.22     |
> >| DoRA (QKV) $r=16$               | 1.51                     | 70.40     | 90.31     | 86.03     | 57.81     | 86.77     | 93.92     | 89.30     | 92.48     |
> >| Full Fine-tune (QV) $\lambda=2$ | 14.76                    | 73.53     | _91.01_   | 86.02     | 60.57     | 62.03     | 93.11     | **90.56** | 91.96     |
> >| Full Fine-tune (QV) $\lambda=4$ | 14.76                    | 72.29     | 90.56     | 87.01     | **61.88** | 35.44     | 91.05     | 89.81     | 88.85     |
> >| Full Fine-tune (QV) $\lambda=8$ | 14.76                    | 72.29     | 90.02     | _88.97_   | _61.86_   | 35.44     | 84.75     | 85.93     | 50.54     |
> >| LoRA (QV) $r=8, \lambda=2$      | 1.48                     | 71.84     | 90.37     | 86.02     | 58.54     | 86.85     | 94.03     | 89.47     | 92.33     |
> >| LoRA (QV) $r=8, \lambda=4$      | 1.48                     | 75.09     | 90.83     | 87.01     | 59.56     | 86.95     | 94.04     | 90.09     | 92.86     |
> >| LoRA (QV) $r=8, \lambda=8$      | 1.48                     | 76.13     | 90.75     | _88.97_   | **61.88** | 86.93     | 93.46     | 90.01     | 92.34     |
> >| LoRA (QV) $r=16, \lambda=2$     | 1.77                     | 70.39     | 90.46     | 86.03     | 58.55     | 86.83     | **94.38** | 89.77     | 92.33     |
> >| LoRA (QV) $r=16, \lambda=4$     | 1.77                     | _76.17_   | **91.05** | 87.99     | 60.06     | _87.19_   | 94.03     | 90.30     | 92.73     |
> >| LoRA (QV) $r=16, \lambda=8$     | 1.77                     | 72.92     | 90.96     | **89.95** | 59.31     | **87.31** | 93.92     | 90.43     | 92.95     |
> >| DoRA (QV) $r=8, \lambda=2$      | 0.90                     | 71.12     | 90.29     | 87.01     | 58.54     | 87.08     | 93.96     | 89.60     | 92.60     |
> >| DoRA (QV) $r=8, \lambda=4$      | 0.90                     | 75.45     | 90.82     | 86.76     | 60.32     | 86.98     | 93.81     | 90.33     | _92.97_   |
> >| DoRA (QV) $r=8, \lambda=8$      | 0.90                     | 70.76     | 90.38     | 87.75     | 57.01     | 87.12     | 94.15     | 90.45     | 92.48     |
> >| DoRA (QV) $r=16, \lambda=2$     | 1.20                     | 69.68     | 90.53     | 87.75     | 59.31     | 87.09     | 93.92     | 89.68     | 92.70     |
> >| DoRA (QV) $r=16, \lambda=4$     | 1.20                     | 76.16     | 90.77     | 88.48     | 60.84     | 86.96     | 94.15     | 90.34     | **93.01** |
> >| DoRA (QV) $r=16, \lambda=8$     | 1.20                     | **77.26** | 90.83     | 88.96     | 60.32     | 87.10     | _94.17_   | 90.46     | 92.80     |

---

> ### Author Response · Authors · 2024-11-16
>
> >**W1:** **Generalizability of the Proposed Approach**: The primary concern is the generalizability of the proposed approach. Specifically, the authors could enhance the analysis by demonstrating that the proposed method consistently improves LLM performance across diverse scenarios. To this end, it would be beneficial to include performance results under more complex, open-ended generation tasks, such as MT-Bench or comparable challenging benchmarks. Additionally, considering variations in model behavior, evaluating the approach on a broader range of LLMs, such as Mistral, would further strengthen the claim of general applicability.
>
> >**A:**  Good suggestions !
> >
> >(1) In alignment with the experimental setup (hyperparameter setting for Llama3.1-8b) described in our Section 4.1, Figure 2, we have evaluated the RTE and MNLI task performances of our approach on Mistral-7B (**you mentioned**). Below are the results from this additional analysis:
> >
> >($\eta_{QK}=1e^{-5},\eta_V=\lambda\eta_{QK}$)
> >
> >| **Method** | Hyperparameter        | RTE       | MNLI      |
> >| ---------- | --------------------- | --------- | --------- |
> >| LoRA (QKV) | $r=16$,$\lambda=1$   | 81.28     | 87.80     |
> >| LoRA (QV)  | $r=16$,   $\lambda=1$ | 80.51     | 88.87     |
> >| LoRA (QV)  | $r=16$,   $\lambda=2$ | 81.59     | **89.04** |
> >| LoRA (QV)  | $r=16$,   $\lambda=4$ | 80.87     | 88.64     |
> >| LoRA (QV)  | $r=16$,   $\lambda=8$ | **83.75** | 88.78     |
> >
> >(2) Given that our work is on the theoretical analysis of attention mechanisms during fine-tuning and improving various fine-tuning methods, and MT-Bench [1] is designed as an evaluation benchmark dataset (the dataset contains only 80 samples), could you kindly elaborate on why you recommend this specific task? Are you suggesting that we fine-tune and evaluate our approach on this benchmark, or is there a particular aspect of MT-Bench that you believe would align with our theoretical contributions? Your clarification would help us better understand and address your suggestion.
> >
> >[1] https://github.com/lm-sys/FastChat/tree/main/fastchat/llm_judge#mt-bench
>
> >**W2:**  **Visualization for Better Insight**: To deepen the analysis and understanding of the findings, visualizing the learned attention distributions across different settings could be valuable. By examining how attention distributions vary under different configurations, the authors could offer a more nuanced understanding of the observed effects, shedding light on the underlying phenomenon.
>
> >**A:** We appreciate the suggestion to visualize attention distributions to deepen the analysis. However, due to the complexity of the model architecture (e.g., multi-layer and multi-head attention) and the variability of attention scores across different input data, it may be challenging to establish a standardized evaluation metric for determining the quality of the visualized attention distributions.
>
> We wonder if we might have misunderstood your suggestion. Could you kindly clarify or provide further details? Your guidance would greatly help us in addressing your point effectively and improving our work.

---

> ### Author Response · Authors · 2024-11-26
> **We are looking forward to hearing from you！**
>
> Dear Reviewer SWzh,
>
> We sincerely appreciate your time and effort in reviewing our manuscript and providing valuable feedback.
>
> As the day that authors may upload a revised PDF phase nears completion, we wish to confirm whether our responses have effectively addressed your concerns. We provided detailed responses to your concerns a few days ago and hope they have adequately resolved any issues. If you require further clarification or have any additional concerns, please do not hesitate to contact us. We are more than willing to continue our communication with you.
>
> Best regards.

---

> ### Author Response · Authors · 2024-11-30
> **We eagerly await your response!**
>
> Dear Reviewer SWzh,
>
> We are eager to know if you have any additional concerns or suggestions regarding our paper. If there are none, we sincerely hope you might consider raising the review score, and we would be more than willing to engage in detailed technical communication to address any potential concerns.
>
> Best regards.

---

### Author Response · Authors · 2024-11-23
**Global Response (part1)**

Dear everyone,

We sincerely appreciate the detailed and constructive feedback provided by all reviewers. We are encouraged to note that the reviewers have recognized:

>- Reviewer SWzh: This paper offers both **empirical and theoretical analyses** to elucidate the behavior of the attention mechanism during fine-tuning. **The insights provided here could make a valuable contribution to the field, supporting the development of enhanced fine-tuning techniques that optimize the accuracy-efficiency trade-off**.
>- Reviewer FTN2: The authors employ **both information-theoretic methods and optimization theory** to support their claims, which significantly strengthens the credibility of their contributions..... This contribution is highly relevant for **both academia and industry**, as it provides a more resource-efficient approach to adapting pre-trained models for downstream tasks.
>- Reviewer f8uV: The idea of using different local learning rates for Wq and Wv is interesting. As V is timed with attention logits, q and v should have more different gradient distributions than q and k. **Using different local learning rates for these two types of matrices is reasonable and worth exploring.**
>- Reviewer TXeN: The observation that fine-tuning only Wq and Wv matrices yields performance gains comparable to finetuning of Wq, Wk and Wv is interesting. **Trying to analyze and uncover the reason can benefit the research commnity**.
>
We have addressed all the questions raised by the reviewers through detailed clarifications, providing separate responses to each reviewer (Following **Reviewer f8uV's suggestion**, we use colored text to show the modifications and improvements in the rebuttal revision). Additionally, we would like to address some common concerns in a consolidated global response.

>**(1) The main contribution of this work.**
>
>Our primary objective is to provide a general **theoretical framework** that reveals the underlying mechanism behind the  phenomena observed during the fine-tuning of LLMs involving the attention mechanism. Building on our experimental and theoretical insights, **one can develop new algorithms** to improve the effectiveness (e.g., storage, and time) of fine-tuning.
>
> According to the traditional statistical learning viewpoint, performance can be defined by the sum of **optimization error and generalization error**. Our theoretical analyses in Sections 3 and 4 correspond to generalization and optimization, respectively.
>
>- In Section 3 (generalization), we give **Thereom 1** (Information-theoretic genralization bounds), showing that with the same r value, fine-tuning $W_q,W_v$ consistently achieves results comparable to or even surpassing those of fine-tuning $W_q,W_k,W_v$. This reduces the number of parameters for the same r, while improving generalization bounds and potentially providing memory benefits.  (**storage-friendly**)
>
>- In Section 4 (optimization), we discuss the learning dynamics in fine-tuning attention mechanism, and we illustrate (**Theorem 2**) that the feature learning of attention mechanism is efficient when the learning rate for $W_v$ should be generally much larger than that of $W_q,W_k$ in fine-tuning. (**time-friendly**)
>
>Building on our experimental and theoretical insights, example in Section 5 is presented solely to illustrate how theoretical analysis can guide experimental procedures effectively.

>**(2) How to choose good $\lambda$ values from both theoretical and empirical aspects?**
>
>Theoretically:
>
>As discussed in **Remark 4**, the optimal ratio $\lambda$ depends on the architecture and the fine-tuning task via the constants in $\Theta$ in **Theorem 2**.  This is a limitation of these asymptotic results since they do not offer any insights on how the constants are affected by the task and the neural architecture.
>
>Empirically:
>
>However, we can still employ some heuristic methods, such as: we can select an appropriate range by conducting a certain amount of experiments, as shown in **Figure 1**, it seems that a ratio of order $2^1-2^4$ is optimal. （That is why we do a simple search for $\lambda$ (2,4,8) in Section 5.）
>
>Moreover, $\lambda$ should not be too large; otherwise, as shown in the MNLI subplot in Figure 1, the model's performance will collapse ($\eta_{QK}=4e^{-4},\eta_V=2e^{-3}$).

---

> ### Author Response · Authors · 2024-11-23
> **Global Response (part2)**
>
> >**(3) The ablation studies presented in the main text**.
> >
> >(i) For the effects of different components ( such as $W_q$ vs. $W_v$ ).
> >
> >In **Section 3.1 (Table 1)**, we provide a detailed comparison of the impact of fine-tuning different weight matrices.
> >
> >We can see a clear trend where solely updating the $W_v$ matrix outperforms just learning the $W_q,W_k$ matrix. Interestingly, the combination of fine-tuning both $W_q$ and $W_v$ often leads to performance that matches or even exceeds that achieved by fine-tuning all three matrices $W_q,W_k,$ and $W_v$. This pattern is consistently observed across various tasks and rank values, further emphasizing the importance of these two matrices over $W_k$ during fine-tuning.
> >
> >(ii) For the effect of learning rate scaling.
> >
> >In **Section 4.1 (Figure 1 and Figure 2)**, we compared the results of various learning rate combinations ($\eta_V,\eta_{QK}$).
> >
> >We observe that (a) test accuracy consistently reaches its maximum for certain sets of learning rates where $\eta_{QK}< \eta_V$,  outperforming the standard practice of setting $\eta_{QK}$ and $\eta_V$ equal. (b) More interestingly, the gap between the optimal choice of learning rates overall and the optimal choice when $\eta_{QK}=\eta_V$ varies across different tasks.
>
> >**(4) For more ablations to demonstrate.**
> >
> >We have added ablation experiments on different fine-tuning methods (Dora [1]) and models (Mistral-7B [2]). Please refer to our supplementary experiments for details.
> >
> >Following the experimental setup (hyperparameter setting for Llama3.1-8b) described in our Section 5, Table 2, we conducted additional evaluations on DoRA: (Revision line 493- line 510)
> >
> >>| **Method**                  | **Trainable #Param (M)** | **RTE**   | **STS-B** | **MRPC** | **CoLA**  | **MNLI**  | **SST-2** | **QQP**   | **QNLI**  |
> >>| --------------------------- | ------------------------ | --------- | --------- | -------- | --------- | --------- | --------- | --------- | --------- |
> >>| DoRA (QKV) $r=8$            | 1.06                     | 70.75     | 90.39     | 85.78    | 56.79     | 86.73     | 93.58     | 89.34     | 92.22     |
> >>| DoRA (QKV) $r=16$           | 1.51                     | 70.40     | 90.31     | 86.03    | 57.81     | 86.77     | 93.92     | 89.30     | 92.48     |
> >>| DoRA (QV) $r=8, \lambda=2$  | 0.90                     | 71.12     | 90.29     | 87.01    | 58.54     | 87.08     | 93.96     | 89.60     | 92.60     |
> >>| DoRA (QV) $r=8, \lambda=4$  | 0.90                     | 75.45     | 90.82     | 86.76    | 60.32     | 86.98     | 93.81     | 90.33     | _92.97_   |
> >>| DoRA (QV) $r=8, \lambda=8$  | 0.90                     | 70.76     | 90.38     | 87.75    | 57.01     | **87.12** | 94.15     | 90.45     | 92.48     |
> >>| DoRA (QV) $r=16, \lambda=2$ | 1.20                     | 69.68     | 90.53     | 87.75    | 59.31     | 87.09     | 93.92     | 89.68     | 92.70     |
> >>| DoRA (QV) $r=16, \lambda=4$ | 1.20                     | 76.16     | 90.77     | 88.48    | **60.84** | 86.96     | 94.15     | 90.34     | **93.01** |
> >>| DoRA (QV) $r=16, \lambda=8$ | 1.20                     | **77.26** | 90.83     | 88.96    | 60.32     | 87.10     | **94.17** | **90.46** | 92.80     |
> >
> >In alignment with the experimental setup (hyperparameter setting for Llama3.1-8b) described in our Section 4.1, Figure 2, we have evaluated the RTE and MNLI task performances of our approach on Mistral-7B :
> >
> >| **Method** | Hyperparameter        | RTE       | MNLI      |
> >| ---------- | --------------------- | --------- | --------- |
> >| LoRA (QKV) | $r=16$，$\lambda=1$   | 81.28     | 87.80     |
> >| LoRA (QV)  | $r=16$,   $\lambda=1$ | 80.51     | 88.87     |
> >| LoRA (QV)  | $r=16$,   $\lambda=2$ | 81.59     | **89.04** |
> >| LoRA (QV)  | $r=16$,   $\lambda=4$ | 80.87     | 88.64     |
> >| LoRA (QV)  | $r=16$,   $\lambda=8$ | **83.75** | 88.78     |
> >
> >[1] Liu, Shih-Yang, et al.  Dora: Weight-decomposed low-rank adaptation.  ICML Oral 2024.
> >
> >[2] https://github.com/mistralai
>
> Thank you once again to all the reviewers for your patience and invaluable comments. We hope our responses have clarified your initial concerns and questions. We are happy to provide further clarifications if necessary.

---

### Meta-Review · Area_Chair_mwbW · 2024-12-13

**Metareview:**

While the paper addresses an important topic—fine-tuning strategies for attention mechanisms in large language models—and presents some interesting observations, the contributions appear incremental and insufficiently distinct from prior work. The theoretical insights provided lack intuitive explanations and fail to clearly articulate novel contributions beyond known observations in the literature. Furthermore, the paper's writing is frequently unclear. Additionally, the lack of adequate baselines, ablation studies, and broader evaluations on diverse and challenging benchmarks limits the generalizability of the claims.

**Additional Comments On Reviewer Discussion:**

Reviewers raised concerns about the paper's originality, clarity, alignment of claims with experimental evidence, and lack of generalizability. Specific points included similarities to the LoRA paper, inconsistencies between claims and results (e.g., selective fine-tuning often not matching full fine-tuning), unclear theoretical sections, limited evaluations, and insufficient ablations. The authors clarified differences from LoRA, revised writing for clarity, added ablations, and emphasized theoretical contributions. However, key concerns remained, such as incremental contributions, unresolved misalignment between claims and results, and insufficient exploration of broader benchmarks.

---

### Decision · Program_Chairs · 2025-01-22

Reject